# Structure of engineered hepatitis C virus E1E2 ectodomain in complex with neutralizing antibodies

Matthew C. Metcalf[1,2], Benjamin M. Janus [1,2], Rui Yin [1,2], Ruixue Wang[2], Johnathan D. Guest[1,2], Edwin Pozharski [2,3,4], Mansun Law [5], Roy A. Mariuzza [1,2], Eric A. Toth [2], Brian G. Pierce [1,2], Thomas R. Fuerst [1,2] & Gilad Ofek [1,2] ✉

Hepatitis C virus (HCV) is a major global health burden as the leading causative agent of chronic liver disease and hepatocellular carcinoma. While the main antigenic target for HCV-neutralizing antibodies is the membrane-associated E1E2 surface glycoprotein, the development of effective vaccines has been hindered by complications in the biochemical preparation of soluble E1E2 ectodomains. Here, we present a cryo-EM structure of an engineered, secreted E1E2 ectodomain of genotype 1b in complex with neutralizing antibodies AR4A, HEPC74, and IGH520. Structural characterization of the E1 subunit and C-terminal regions of E2 reveal an overall architecture of E1E2 that concurs with that observed for non-engineered full-length E1E2. Analysis of the AR4A epitope within a region of E2 that bridges between the E2 core and E1 defines the structural basis for its broad neutralization. Our study presents the structure of an E1E2 complex liberated from membrane via a designed scaffold, one that maintains all essential structural features of native E1E2. The study advances the understanding of the E1E2 heterodimer structure, crucial for the rational design of secreted E1E2 antigens in vaccine development.

An estimated 58 million people are infected with hepatitis C virus (HCV)[1]. Approximately 75% of infections become chronic and in turn can lead to cirrhosis or hepatocellular carcinoma, a major cause of liver-related deaths. The development of direct-acting antivirals provides a possible cure for infection but does not prevent reinfection. Additionally, direct-acting antiviral treatment is inaccessible for the majority of the infected population, with less than 10% of the global population having access to treatment due to healthcare and financial constraints. These limitations suggest that the most viable method for controlling HCV infections worldwide is through the development of a prophylactic vaccine[2–4].

Although HCV is classified as a flavivirus, its two surface envelope glycoproteins, E1 and E2, do not appear to share sequence or structural features with other flavivirus type II fusion glycoproteins[5,6]. Both E1 and E2 are single-pass transmembrane proteins with N-terminal ectodomains of 160 and 330 residues, respectively, and C-terminal transmembrane domains of roughly 30 residues. E1 and E2 form heterodimers on the surface of the virion and have been proposed to form higher-order trimers, although a recent structure of membrane-extracted E1E2 heterodimer did not reveal higher-order oligomers[7–10]. Both E1 and E2 subunits are heavily glycosylated, with E1 containing 5 to 6 predicted N-linked glycosylation sites and E2 between 9 to 11,

[1]Department of Cell Biology and Molecular Genetics, University of Maryland, College Park, MD, USA. [2]Institute for Bioscience and Biotechnology Research, University of Maryland, Rockville, MD, USA. [3]Center for Biomolecular Therapeutics, University of Maryland School of Medicine, Baltimore, MD, USA. [4]Department of Biochemistry and Molecular Biology, University of Maryland School of Medicine, Baltimore, MD, USA. [5]Department of Immunology and Microbiology, The Scripps Research Institute, La Jolla, CA, USA. ✉e-mail: gofek@umd.edu

depending on the genotype. Numerous conserved cysteines within both subunits mediate extensive disulfide bonding networks and have been suggested to be essential for viral entry but not for heterodimerization[11,12]. The E2 subunit interacts with various cellular entry receptors, including tetraspanin CD81, scavenger receptor-B1, occludin, and claudin. The E1 subunit has been presumed to play a role in the fusion of the viral and host membranes during entry, although sequence homology to other membrane fusogens is difficult to discern[13–15].

Multiple studies have identified broadly neutralizing antibodies (bnAbs) against HCV both from acute and chronic infection, with the former being associated with viral clearance[16]. Epitope mapping and structural characterization of such bnAbs have defined antigenic regions (AR) or antigenic domains that they target, which fall on the individual E1 or E2 subunits or depend on E1E2 heterodimers for recognition[5–7,16–22]. Varying nomenclatures have been used to describe these sites, including AR 1-5, antigenic domains A-E, and epitopes I-III. AR3 or antigenic domains B, D, and E on E2, collectively known as the E2 neutralizing face, are highly conserved and overlap with the CD81 host receptor binding region[16,23]. AR4 and AR5 outside the receptor binding region are also targets of bnAbs, but ones that depend on an intact E1E2 heterodimer for recognition. These include antibodies AR4A and AR5A, as well as antibodies HEPC111 and HEPC130[16,18,22]. While heterodimer-specific bnAbs do not prevent CD81 binding, they are among the most broadly neutralizing antibodies identified to date[18].

Structural characterization of antigenic targets on E2 has provided a great deal of information on sites of immune vulnerability on the virus, notably for bnAb-targeting epitopes overlapping the CD81 receptor binding domain[20,24–29]. Development of an HCV vaccine has nonetheless proven challenging due to multiple factors[2,16,30]. These include a high degree of E1E2 genetic diversity (7 genotypes and over 90 subtypes), conformational plasticity, glycan shielding of neutralizing epitopes, presence of immunodominant non-neutralizing epitopes, and a proclivity for aggregation[2,31–35]. Past efforts to design E1E2-based vaccine antigens have included membrane-extracted full-length E1E2s with intact transmembrane domains, covalent E1E2 ectodomain fusions, and soluble scaffolded E1E2 ectodomain approaches[36–39]. While a majority of designed immunogens failed to produce E1E2 proteins that recapitulated native antigenic recognition by E1E2 heterodimer-specific antibodies, membrane-extracted full-length E1E2s generally preserved such recognition. Indeed, use of a full-length membrane-extracted E1E2 antigen in an early-stage human phase 1a clinical trial resulted in the induction of neutralizing antibodies, although broad cross-neutralization was only observed in one of 16 participants[40,41]. Recently, a cryo-EM structure of a full-length membrane-extracted E1E2 of genotype 1a was reported in complex with a set of antibodies that included heterodimer-specific antibody AR4A[7]. The structure revealed a number of features of the native E1E2 heterodimer, including the overall assembly of E1E2, the interface between E1 and E2, recognition of E1E2 by neutralizing antibodies, and the orientation of E1E2 relative to the viral membrane[7]. While such full-length membrane-extracted forms of E1E2 retain native antigenic recognition, biochemical production challenges that result from the presence of intact transmembrane domains render such forms of E1E2 less amenable to vaccine development as compared to E1E2 antigens that are liberated from membrane, if the latter properly mimic the native E1E2 structure.

We previously engineered a class of soluble, secreted E1E2 (sE1E2) heterodimer ectodomains that untethered E1E2 from membrane by replacing the native E1 and E2 transmembrane domains with leucine zipper scaffolds, and that maintained native E1E2 heterodimerization by inclusion of a furin cleavage site between E1 and E2[42]. Recognition of sE1E2 proteins by heterodimer-specific bnAbs such as AR4A and AR5A indicates they retain native E1E2 antigenic integrity[42]. Due to their soluble nature, sE1E2s offer the advantages of uniform production, ease of biochemical characterization, and ease of quality control, and thus overall utility as vaccine immunogens. They also offer greater tractability for structural analysis for the very same reasons.

Here, we describe a 3.65 Å resolution cryo-EM structure of a coiled coil-scaffolded sE1E2 heterodimer, sE1E2.SYNZIP (or sE1E2.SZ for short), of genotype 1b in complex with neutralizing antibodies AR4A, HEPC74, and IGH520. The obtained structure delineates the heterodimeric architecture of the E1 and E2 subunits free of membrane association, including regions of E1 and the C-terminus of E2 that comprise the E1E2 interface. Sequence analysis of E1E2 across HCV genotypes reveals the AR4A epitope to be among the most highly conserved. Comparison of the sE2E2.SZ structure to that of the full-length non-engineered membrane-extracted E1E2 of genotype 1a reveals a high degree of structural homology, confirming that sE1E2.SZ retains native structural features desired for vaccine development[7].

## Results

### Engineered HCV E1E2 for structural analysis

An sE1E2 ectodomain construct, sE1E2.SZ, of HCV genotype 1b isolate 1b09 was engineered as a polyprotein expression construct based on a previously described strategy in which the transmembrane domains of E1 and E2 were individually substituted with leucine zipper forming coiled coils[42]. To avoid potential immunogenic complications related to the induction of immune responses against native leucine zippers in humans, we utilized a modified synthetic leucine zipper scaffold, SYNZIP, composed of coils SYNZIP1 and SYNZIP2 that self-assemble into stable coiled-coils[42,43]. The SYNZIP1 coil was fused to the C-terminus of E2 and the SYNZIP2 coil to the C-terminus of E1, in each case replacing the respective native transmembrane domains (Fig. 1a). A furin cleavage site was introduced at the C-terminus of SYNZIP2 to ensure proteolytic processing of the sE1E2.SZ polyprotein and native heterodimerization (Fig. 1a). The wild-type signal peptide sequence was replaced with an engineered signal sequence of tissue plasminogen activator (tPA) to ensure efficient secretion.

### Quinary complex of sE1E2.SZ with antibodies AR4A, HEPC74, and IGH520

While previous studies of sE1E2 engineered with leucine zipper scaffolds yielded robust expression in transient mammalian expression systems, their relatively small size and polydispersity hampered cryo-EM analysis[42]. We therefore proceeded with an expression and purification strategy that involved co-expression and co-purification of sE1E2.SZ with E1E2 heterodimer-specific bnAb AR4A with the goal of increasing the yield of purified monodisperse sE1E2.SZ and the effective size of the heterodimer through the bound antibody ligand[18]. Analogous strategies have been shown to be effective for the purification of other metastable viral glycoproteins, including RSV F, HCV E2 monomers, and, more recently, full-length membrane-extracted HCV E1E2[7,18,44]. An expression construct for the AR4A Fab fused to a Strep-II tag was synthesized and transiently co-expressed with sE1E2.SZ in Expi293F cells that lack N-acetylglucosaminyltransferase I (GnTI) activity and therefore produced protein without complex N-linked glycans. Streptactin resin was used to purify co-expressed sE1E1.SZ-AR4A complex, ensuring only AR4A-bound sE1E2.SZ heterodimers were isolated and separated from the remaining, less monodisperse population of sE1E2.SZ (Supplementary Fig. 1a, b). To further increase the molecular size and facilitate cryo-EM analysis, we generated a higher-order quinary complex of the purified sE1E2.SZ-AR4A complex by combining it with two additional Fabs: first with IGH520, which targets a contiguous epitope within the C-terminal region of E1 (Supplementary Fig. 1c, d), and then with HEPC74 that targets the E2 neutralizing face[45,46] (Supplementary Fig. 1e, f). The presence of all five components of the quinary complex was confirmed by size exclusion chromatography and SDS-PAGE analysis, and the resulting complex

was concentrated and used for cryo-EM grid preparation (Supplementary Fig. 1).

## Cryo-EM data collection and processing

The prepared sE1E2.SZ-AR4A-HEPC74-IGH520 complex was imaged on an FEI Glacios 200 kV TEM equipped with a Gatan K3 camera, and data were collected and processed using SerialEM and cryoSPARC, respectively (Supplementary Fig. 2)[47,48]. Blob picking was implemented on a small subset of the dataset, and the resulting particles were used for 2D classification. Particles corresponding to 2D classes of the E1E2-Fab complexes were used for ab initio reconstruction, and the resulting volume was used to generate 2D templates for a template particle picking job applied to the full dataset. After selecting 2D classes consisting of E1E2-Fab particles, multiple rounds of 3D classification were performed, and classes containing the most density for E1 were selected (Supplementary Fig. 2a–c). These particles were then refined using non-uniform refinement to yield cryo-EM maps with a global resolution of 3.65 Å determined from gold-standard FSC curves

(Supplementary Fig. 2d, f and Supplementary Table 1). Local resolution analysis revealed differences in resolution across the structure (Supplementary Fig. 2e).

An alternate strategy was also undertaken in order to obtain more extensive IGH520 and E1 density (Supplementary Fig. 3). In this approach, 2D and 3D classes containing visible IGH520 signals were selected (Supplementary Fig. 3a–c). After multiple rounds of heterogeneous refinement, particles within this stack underwent particle subtraction in which the Fc regions of the HEPC74 and AR4A Fabs were removed. This was then followed by local refinement using a mask encompassing the remainder of the complex (Supplementary Fig. 3h–k). Merging of the results of the primary map and the local refinement map yielded more extensive maps of IGH520 and E1, although at lower resolution (Supplementary Fig. 4).

## Overall structure of sE1E2.SZ antibody complex

The primary map enabled the determination of a structural model of the sE1E2.SZ complex that included E2 residues 421–704, E1 residues

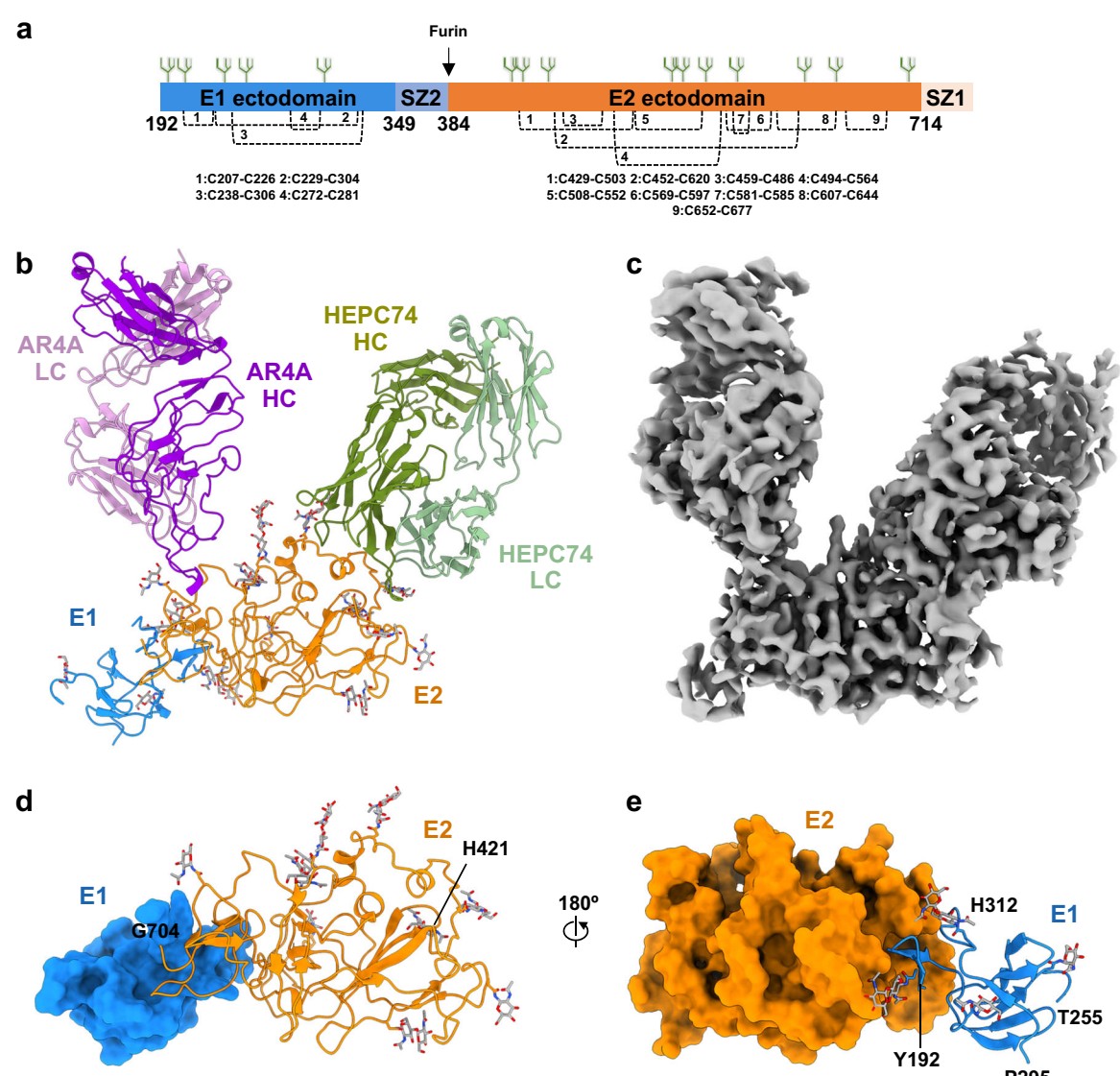

**Fig. 1 | Cryo-EM structure of HCV sE1E2.SZ ectodomain in complex with antibodies. a** Schematic representation of sE1E2.SZ heterodimer ectodomain. Respective leucine zipper SYNZIP (SZ) scaffolds, SZ1 and SZ2, are labeled along with disulfide bonds, N-linked glycans, and the furin cleavage site. **b** Overall cryo-EM structure of sE1E2.SZ (E1, blue; E2, orange) bound by antibodies AR4A (heavy chain,

purple; light chain, pink) and HEPC74 (heavy chain, green; light chain, light green). HC heavy chain, LC light chain. **c** 3.65 Å cryo-EM map used for structure determination. **d** Structure of E2 (orange) shown against a surface representation of E1 (blue). **e** Structure of E1 (blue) rotated by 180° relative to (**d**) and shown against a surface representation of E2 (orange). N-linked glycans are colored gray.

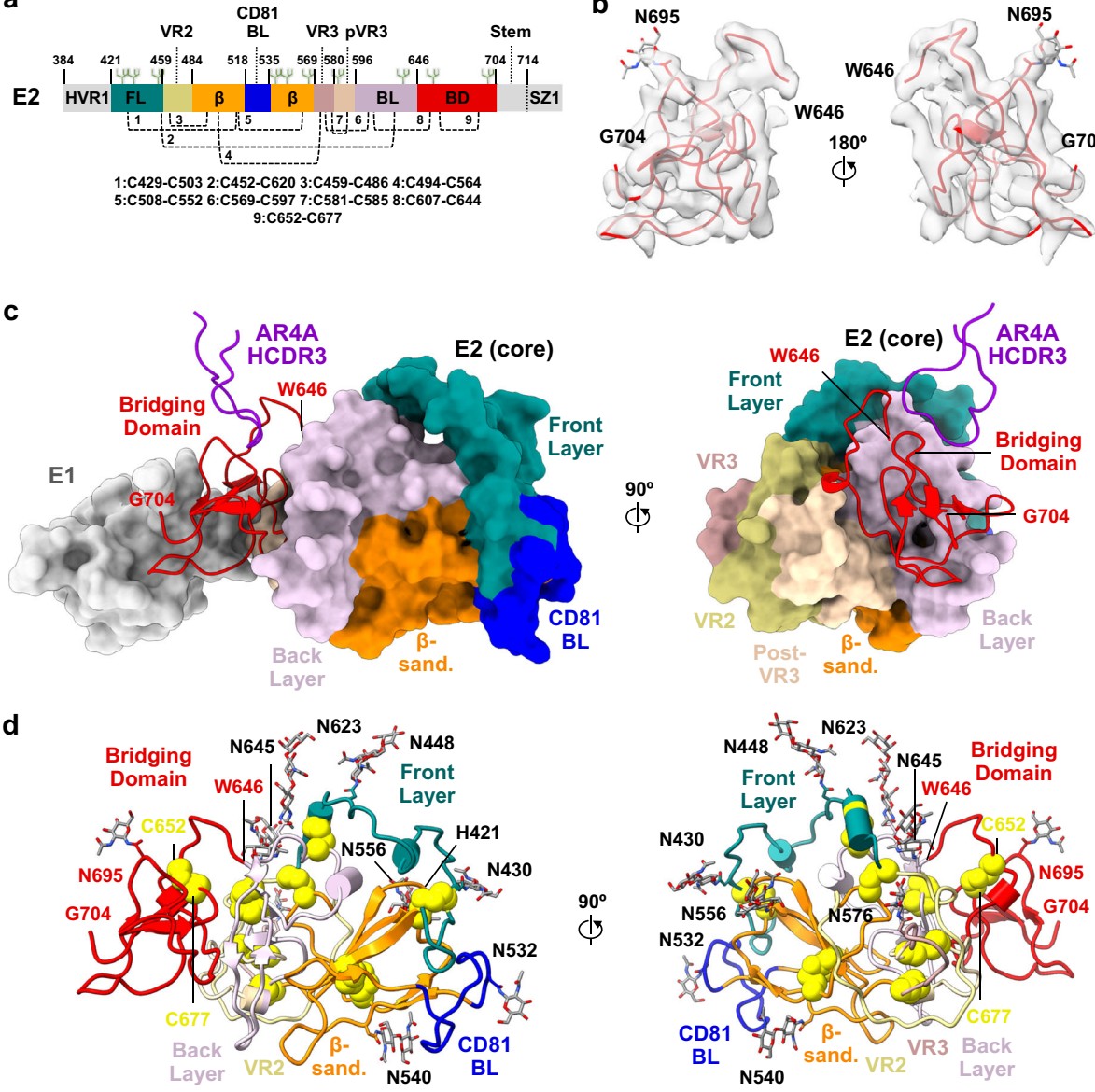

**Fig. 2 | Structural features of E2. a** Schematic representation of E2 colored by subregion. Listed subregions include the front layer (FL, residues 420–458), variable region 2 (VR2, residues 459–483), β-sandwich core (residues 484–517 and 535–568), CD81 binding loop (CD81 BL, residues 518–534), variable region 3 (VR3, residues 569–579), post-variable 3 region (pVR3, residues 580–595), back layer (BL, residues 596–645), and the bridging (or base) domain (BD, residues 646–701). **b** Cryo-EM density for bridging domain, shown in 180° orientations. **c** Surface representation of E2 colored by subregion, shown in 90° orientations. The bridging domain and AR4A HCDR3 loop apex are shown in cartoon representation, red and purple, respectively, with surface representations of E1 shown in gray (left). **d** Cartoon representation of E2 colored by subregion as in (**c**) (left) and rotated 180° (right). N-linked glycans and disulfides are shown as gray sticks and yellow spheres, respectively.

192–255 and 295–312, and bound Fabs HEPC74 and AR4A (Fig. 1b, c). Structure building was aided by N-linked glycan anchor points, positions of bound antibodies, and predicted individual models of E1 and E2 generated by AlphaFold2[49] (Supplementary Fig. 5). Overall, the structure delineated the architecture of the sE1E2.SZ heterodimer ectodomain and the interface between E1 and E2 (Fig. 1b–e and below). No inter-subunit disulfide bonds were observed, nor were higher-order oligomeric states of E1E2. Bound antibody AR4A interacted exclusively with the E2 subunit, targeting a recently resolved region of E2 (aa 646–701), the bridging domain (or base domain), that formed a bridge between the E2 core and E1[7]. Low-resolution density within the merged composite map corresponding to bound antibody IGH520 Fab was detected but was not of sufficient resolution to permit unambiguous structure determination (Supplementary Fig. 4). Nonetheless, the

position of IGH520 aided in the assignment of glycan N305 located immediately upstream of the IGH520 epitope. No density was detected for the SYNZIP leucine zipper scaffold, likely due to flexibility of the stem regions.

### Structure of E2 within the E1E2 heterodimer

The obtained structure of E2 within the sE1E2.SZ heterodimer spanned residues 421–645 of the E2 core, as well as residues 646–704 that made up the bridging domain (Fig. 2a–d). The cryo-EM map enabled building of carbohydrates at nine predicted N-linked glycosylation sites within E2 (N423, N430, N448, N532, N540, N556, N576, N623, N645) to varying extents, as well as at a previously defined non-canonical N-linked glycan site at amino acid N695 (Fig. 2b, d and below)[7]. Nine disulfide bonds were mapped within E2, including eight which were previously

observed in crystal structures of the E2 core, namely between cysteines 429–503, 452–620, 459–486, 494–564, 508–552, 569–597, 581–585, and 607–644[20]. A ninth disulfide bond was observed between cysteines 652 and 677 within the E2 bridging domain (Fig. 2d). C-terminal residues of the E2 stem (residues 705–714) and the appended SYNZIP scaffold were not resolved in the obtained maps and could not be built.

Comparison of E2 core residues 421–645 within sE1E2.SZ against the previously determined crystal structure of E2 core bound to Fab HEPC74 (PDB 6MEH) revealed a high degree of homology between the two, yielding a Cα root mean square deviation (r.m.s.d.) of 1.02 Å (Supplementary Fig. 6)[20]. As such, the structure of the core of E2 appeared largely conserved across its heterodimeric and monomeric states and was not affected by the presence of the E1 subunit[5,6,20,50]. Alignment of sE1E2.SZ E2 against the corresponding crystal structure of E2 bound to tamarin CD81 receptor (PDB 7MWX) revealed differences within the receptor binding loop, but structural homology otherwise (Supplementary Fig. 7).

The E2 bridging domain within sE1E2.SZ exhibited a number of notable features, including a hinge-like loop between residues 646–652, an intra-domain disulfide bond between residues C652 and C677, and a non-canonical glycan at position N695 (Fig. 2b–d). Residues of the bridging domain packed against both the E2 core and against E1. On E2, the bridging domain mainly packed against the E2 back layer, namely with E2 residues 603–607, 624–632, and 639–645, although interactions with post-VR3 residues 584–594 were also observed (Fig. 2c). E1, in turn, appeared to wrap around the bridging domain, and its interface with the bridging domain accounted for roughly 68% of its overall protein interface with E2 (Fig. 2c). The bridging domain also formed the main antigenic interface for antibody AR4A, mainly with its HCDR3 loop (Fig. 2c and below). The presence of the bridging domain between the E2 core and the E1 subunit and its interactions with AR4A implicates its role in the structural integrity of the heterodimer, consistent with the dependence of AR4A recognition on the presence of both the E2 and E1 subunits.

## Structure of the E1 subunit

Cryo-EM map density corresponding to the E1 subunit in sE1E2.SZ enabled the building of a structural model for E1 that corresponded to two discontinuous residue ranges, spanning residues 192–255 and 295–312 (Fig. 3a–c). The putative fusion peptide-containing region (residues 256–294), C-terminal residues of the E1 stem (residues 313–346), and the appended SYNZIP leucine zipper scaffold were unresolved (Fig. 3a). Construction of the E1 model was aided by anchor points corresponding to bulky N-linked glycans (Fig. 3b, c), by an AlphaFold2-generated model of E1 that provided residue assignments for disulfide bonds and N-linked glycans (Fig. 3d), and by lower resolution cryo-EM maps that revealed the spatial placement of IGH520 (Supplementary Fig. 4). Overall, three subregions were structurally defined within E1: an N-terminal domain (NTD), a core domain, and a C-terminal loop region (CTL) (Fig. 3a–d). The N-terminal domain (NTD) spanned E1 residues 192–205 and consisted of two anti-parallel β-strands that packed against E2 variable region 2 (VR2), post-variable region 3 (pVR3), and bridging domain (Figs. 2c and 4e). An N-linked glycan at position N196 of the NTD was well-ordered and formed direct interactions with E2 (Fig. 3b, c and below). The core domain of E1 spanned residues 206–255 and consisted of a cluster of β-strands (Fig. 3). It contained three predicted N-linked glycosylation sites, two of which – N209 and N250 – yielded observable density. A disulfide bond between cysteines C207 and C226 was also observed. Lastly, the resolved portion of the E1 CTL region spanned residues 295–312 and formed interfaces with both E1 and E2. The CTL contained an ordered N-linked glycan at position 305 that packed against residues D653, L654, and Q655 of the E2 bridging domain (see below). Two disulfides connected the E1 CTL to its core domain, namely between C229 and

C304 and between C238 and C306 (Fig. 3a–d). Although a crystal structure of E1 has previously been reported (PDB 4UOI), with the exception of a secondary structure element within the E1 NTD, little structural homology was detected when compared to the present E1 structure[51].

## The E1E2 interface

The interface between E1 and E2 encompassed an average buried surface area of 1054 Å² and was made up entirely of non-covalent interactions between the two subunits, reinforcing previous reports of the absence of intermolecular disulfide bonds between E1 and E2 (Fig. 4)[52,53]. Within E1, the interface was mainly localized to the NTD and CTL domains, while on E2, the interface was split across several regions, most notably on the bridging domain that accounted for more than 60% of the interface with E1 (Fig. 4a, e). Mapping of electrostatic potential onto the surfaces of E1 and E2 revealed the bulk of interactions between the two subunits were hydrophobic in nature, representing roughly 57% of the total residues buried at the interface (Fig. 4b, e).

Two highly conserved E1 N-linked glycans at positions N196 and N305 contributed to the E1E2 interface and collectively accounted for 295 Å² of the buried interface, or 28% of the total interface (Fig. 4d, e). Glycan N196 contributed 202 Å² to the buried interface, while glycan N305 contributed 93 Å² (Fig. 4d, e). We note that previous mutagenesis studies implicated residues N196 and N305 in proper heterodimer integrity and recognition of E1E2 by heterodimer-specific antibodies, confirming their role in E1E2 interface integrity[18,45,54,55].

To assess the degree of sequence variation for interfacial residues of E1 and E2, we calculated Shannon sequence entropies (on a scale of 0 to 100) at each position of the heterodimer across all HCV genotypes and then mapped this value onto the respective E1 and E2 surfaces and sequence (Fig. 4c, e). Interfacial residues on E1 exhibited a mean scaled sequence entropy of 29.8 (weighted by residue buried surface area), while those on E2 exhibited a mean scaled sequence entropy of 13.4 (weighted by buried surface area). Although the E1 interface exhibited more sequence variation than that of E2, interfacial E1 residues were nonetheless conserved in physicochemical nature across multiple HCV genotypes (Fig. 4e).

## AR4A interactions with E2

While previous alanine-scanning mutagenesis studies identified critical residues for AR4A binding in both E1 and E2, and AR4A binding depended on the presence of both subunits, the sE1E2.SZ structure confirmed, as in the case of full-length E1E2, that the AR4A epitope fell exclusively on the E2 subunit, namely on the E2 bridging domain and back layer (Figs. 5 and 2c)[7,18,56]. The sensitivity of AR4A binding to mutations in E1 indicates the conformation of the E2 bridging domain recognized by AR4A is likely stabilized by the E1E2 interface. This is supported by scanning mutagenesis data that alanine replacement at the non-cysteine E1 residues Y201, T204, N205, D206, or I220 similarly reduced binding of antibody AR5A, another E1E2-specific antibody recognizing an epitope distinct from that of AR4A[56].

Interactions between AR4A and E2 were predominantly mediated by the AR4A HCDR3 loop, which accounted for 89% of the 673 Å² of buried surface that made up the antibody paratope (Fig. 5a–d). Minor interactions between the AR4A HCDR2 loop were also observed, although at a greater distance (Fig. 5a–d). Examination of AR4A interactions with E2 in greater detail revealed that a continuous stretch of residues at the apex of the AR4A HCDR3 inserted into a shallow surface cleft formed by two loops of the E2 bridging domain, spanning residues 646–652 and 693–699 (Figs. 5a, b and 2c). Specifically, AR4A HCDR3 residues G100b, T100c, F100d, and L100e formed mostly hydrophobic interactions within this cleft, aided by putative β-like main chain interactions (Fig. 5b). HCDR3 residue W100f packed against a shelf formed by a bridging domain loop between residues 663–667,

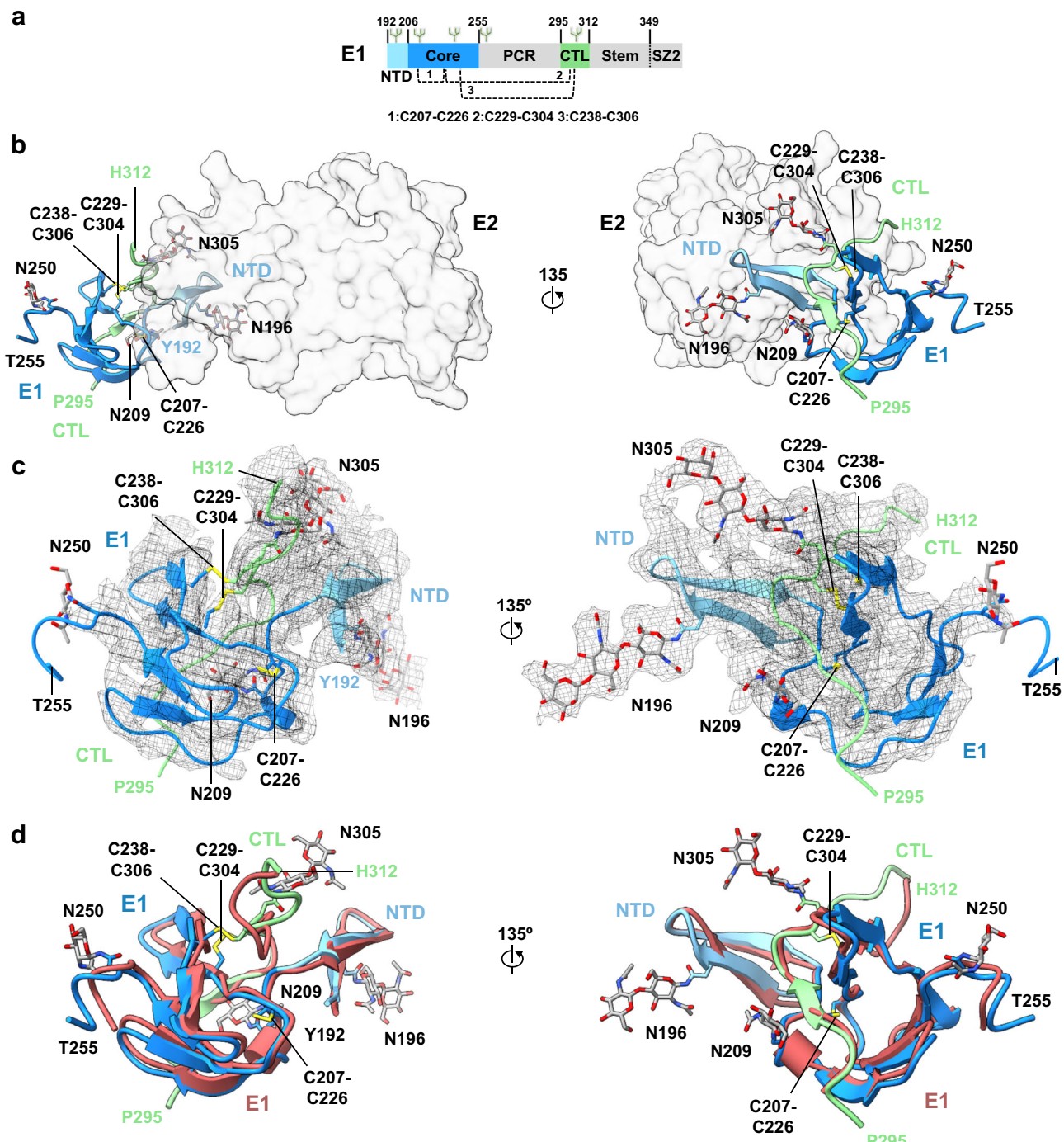

**Fig. 3 | Structural features of E1. a** Schematic representation of E1 colored by subregion. Listed subregions include the N-terminal domain (NTD, residues 192–205), E1 core (residues 206–255 and 295–299), pFP-containing region (PCR, residues 256–298), C-terminal loop (CTL, residues 300–314), and stem (residues 315–351). Regions that were not resolved in the structure are depicted in gray. **b** E1 structure colored by subregion and shown against a semi-transparent surface representation of E2 (light gray), shown in 135° orientations. **c** Cryo-EM map density for E1 structure shown in same orientations as in (**b**). **d** Overlay of E1 structure (colored and oriented as in (**b**)) against an AlphaFold2-generated E1 model (raspberry). N-linked glycans and disulfide bonds are shown as gray and yellow sticks, respectively.

while residues R100g and L100e interacted with the E2 back layer (Fig. 5b). The sole AR4A HCDR2 interaction was between HCDR2 residue Y52i and E2 bridging domain residue R648 (Fig. 5b).

Previous studies have shown that AR4A is one of the broadest neutralizing antibodies against HCV identified to date[57]. To assess the structural basis for broad AR4A recognition, we calculated the mean Shannon sequence entropy across residues of the AR4A epitope on E2, weighted by residue buried surface area, and compared it against

similar values for the epitopes of 13 other HCV E2-directed antibodies, including HEPC74 (Fig. 5e, f). This analysis revealed that the AR4A epitope was among the most highly conserved epitopes examined, with the lowest weighted mean sequence entropy of 16, corresponding to a weighted mean residue conservation of 85% (Fig. 5e)[18,58]. The epitope of HEPC74, in comparison, yielded weighted mean sequence entropy and conservation values of 29 and 70%, respectively (Fig. 5e). Of note, E2 neutralizing face antibodies RM2-01 and RM11-43 were also

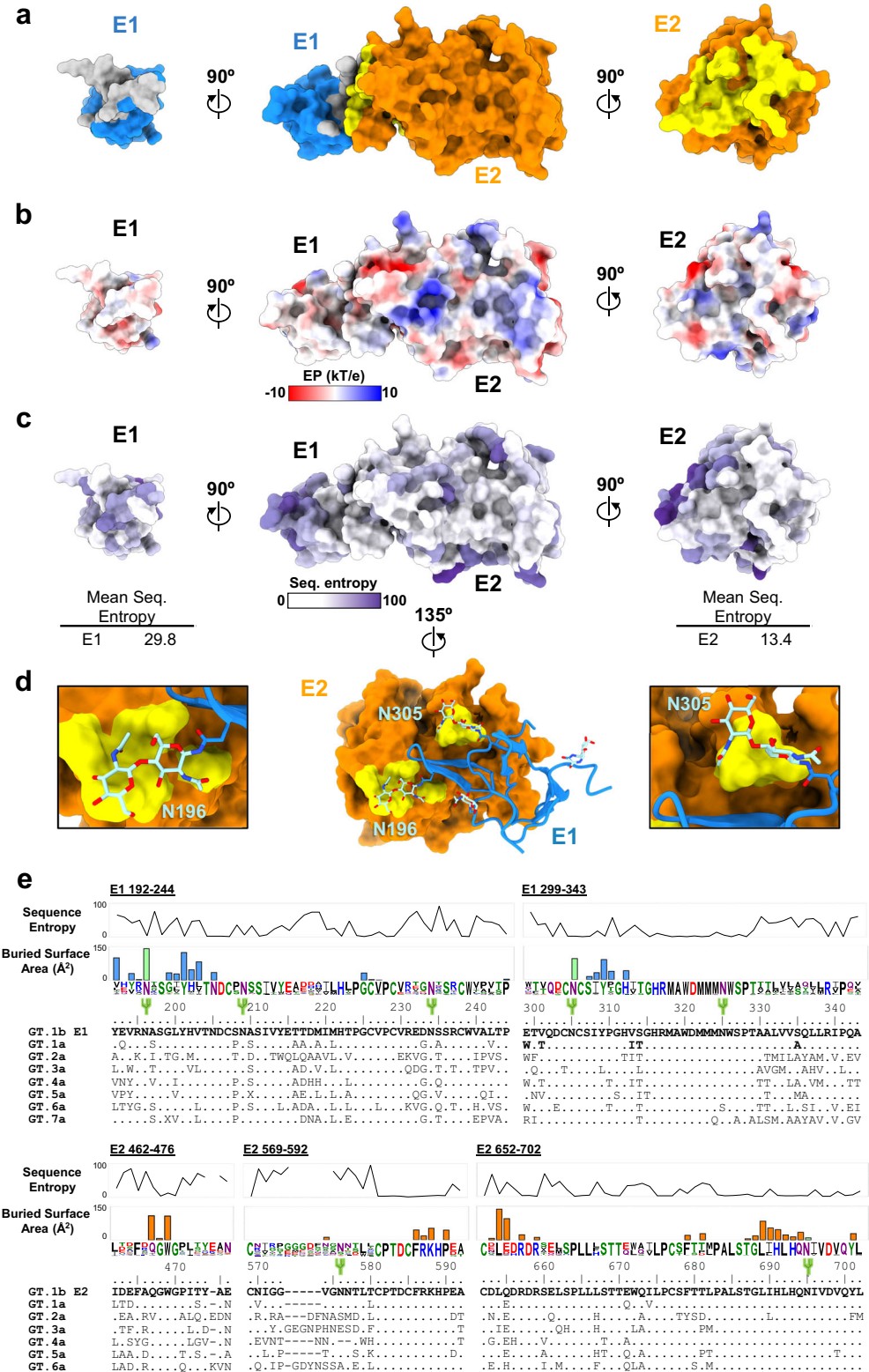

**Fig. 4 | E1E2 interface. a** Surface representation of the sE1E2.SZ heterodimer, middle panel, with 90° open-book views of E1 (left, blue) and E2 (right, orange). E1 and E2 interface residues are colored gray and yellow, respectively. **b** Depiction of the sE1E2.SZ heterodimer as in (**a**), with surfaces colored by electrostatic potential. **c** Depiction of the sE1E2.SZ heterodimer as in (**a**), with surfaces colored by Shannon sequence entropy, scaled on a gradient from 0 (conserved, white) to 100 (variable, purple). **d** E1 N-linked glycans, N196 and N305, that form part of the interface with E2 are shown in cyan sticks along with their respective footprints on E2, yellow. **e** E1 and E2 subregion sequence alignments across HCV genotypes. Sequence Weblogo, scaled sequence entropy, and interface buried surface areas (BSA) at each residue position are plotted above. E1, blue bars; E2, orange bars; BSA contributions of N-linked glycans are shown as light green bars. The GT.1b sequence represents isolate 54-v03 (1b09), which was structurally characterized in this study. Raw sequence entropies were calculated in units of nats. Source data are provided as a Source Data file.

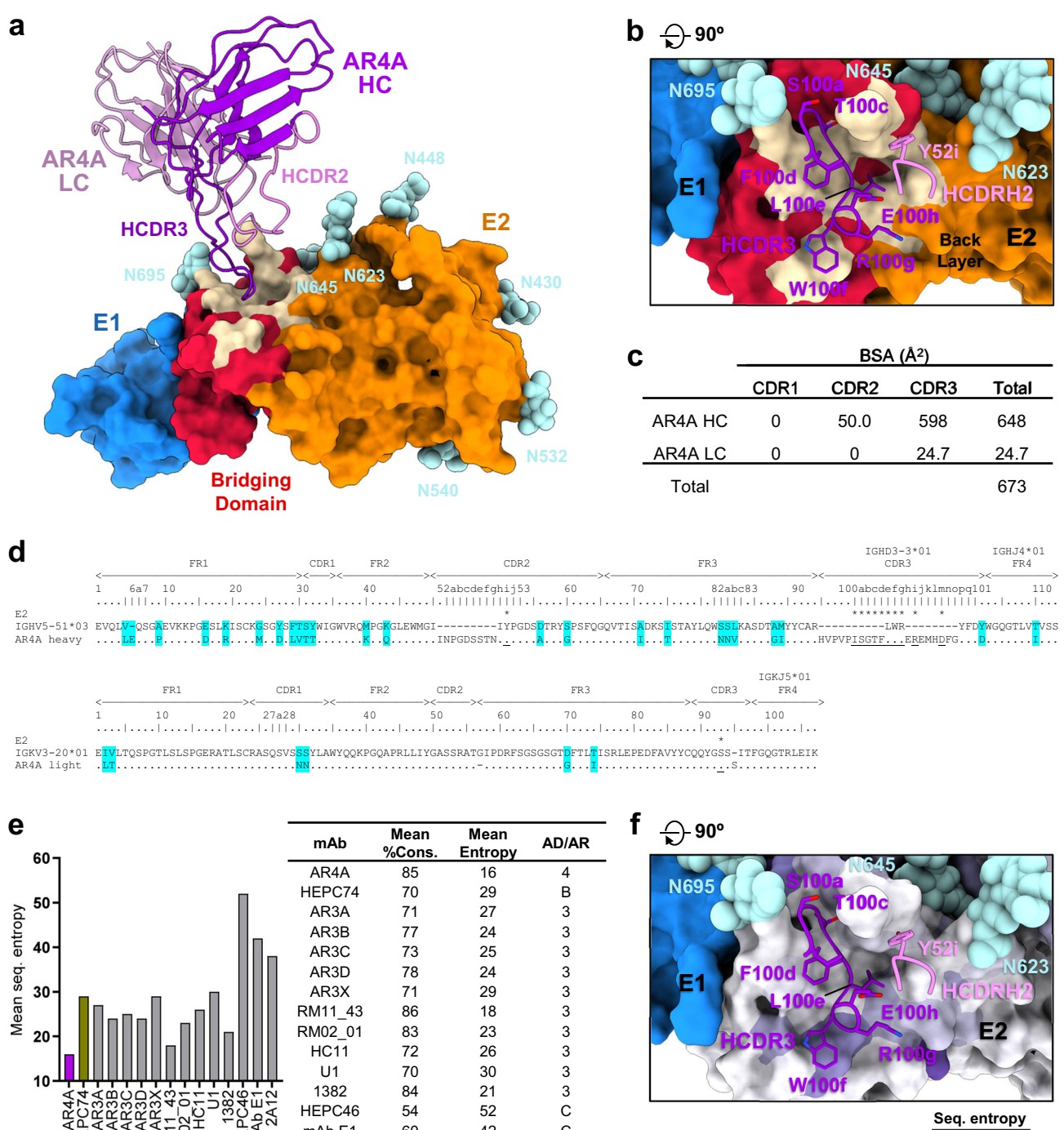

**Fig. 5 | Structural basis for breadth of AR4A recognition. a** Surface representation of sE1E2.SZ bound by AR4A variable region (cartoon). Molecules are colored as in Fig. 1, with the bridging domain colored red and the AR4A epitope on E2 colored wheat. N-linked glycans are shown as cyan spheres. **b** Closeup view of AR4A epitope on E2, rotated by 90° relative to (**a**). AR4A HCDR3 and HCDR2 residues that interact with E2 are shown as sticks and colored purple and magenta, respectively. **c** AR4A heavy chain (HC) and light chain (LC) paratope buried surface areas (BSA), parsed by CDR. **d** Sequence alignment of AR4A heavy and light chains against their respective germline precursors. Residues that interact with E2 are labeled with asterisks and underlined, and somatically matured residues are shaded cyan. **e** Weighted mean scaled Shannon sequence entropy and mean percent sequence conservation of structurally characterized antibody epitopes that target E2 (antigenic domains, AD; antigenic regions, AR). **f** Closeup view of AR4A epitope on E2, rotated by 90° relative to (**a**), with the E2 surface colored by residue sequence entropy on a scale from 0 (conserved, white) to 100 (variable, purple). AR4A residues are depicted as in (**b**). Raw sequence entropies were calculated in units of nats. Source data are provided as a Source Data file.

highly conserved and fell close to the values observed for AR4A (Fig. 5e). The epitopes of antibodies targeting antigenic domains C and A (namely, HEPC46, mAb E1, and 2A12) yielded more pronounced sequence variation with weighted mean sequence entropy and

conservation values that ranged from 38–52 and 54–63%, respectively (Fig. 5e). While the breadth of antibody recognition is influenced both by epitope sequence conservation as well as antibody capacity to tolerate sequence variation, these results nonetheless revealed that

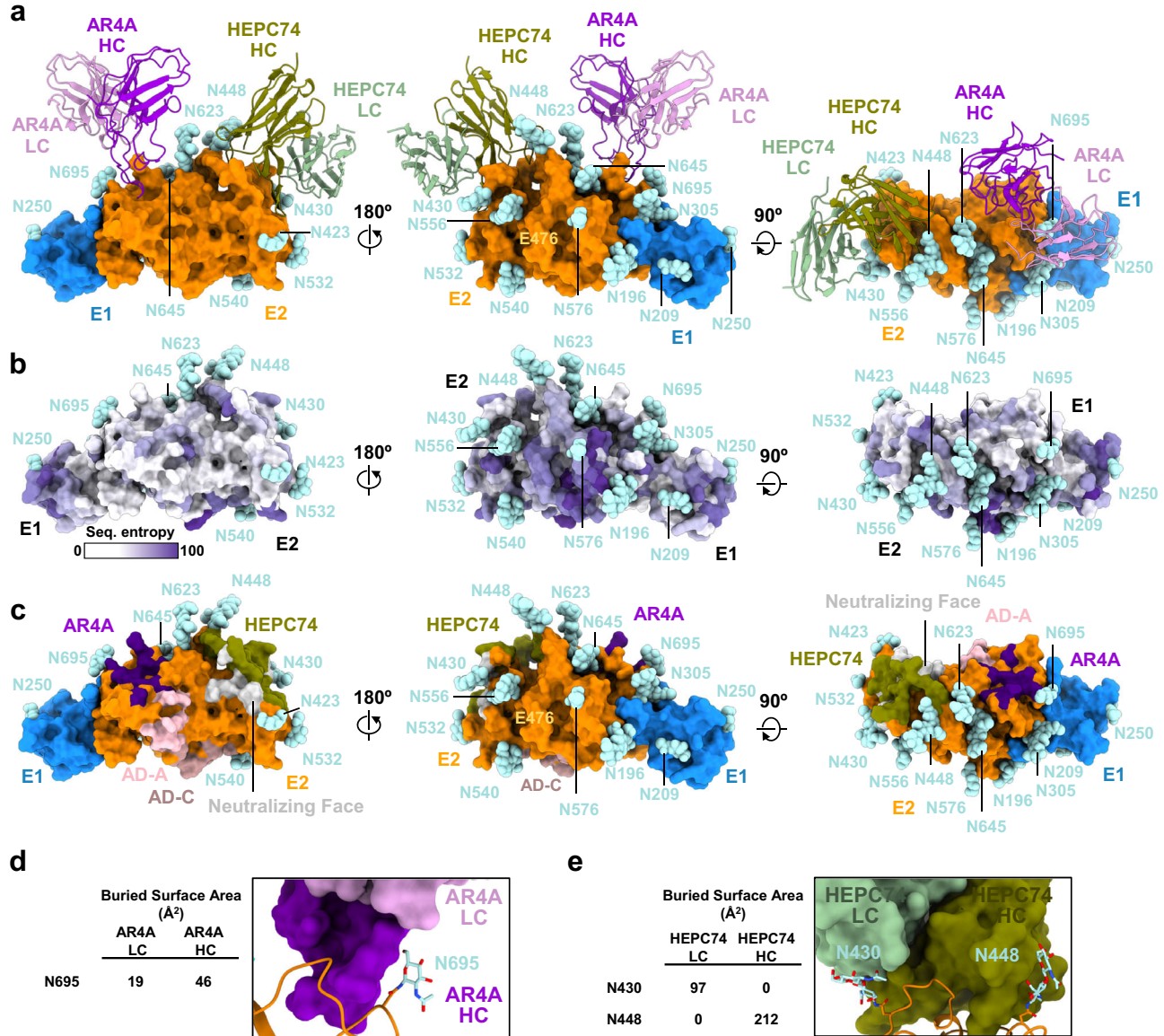

**Fig. 6 | Glycan shield and antigenic targets. a** Surface representation of sE1E2.SZ with bound AR4A and HEPC74 variable regions (cartoon), shown in three orientations and colored as in Fig. 1 with N-linked glycans shown as cyan spheres. **b** Sequence entropy mapped onto the surface of sE1E2.SZ, scaled on a gradient from 0 (conserved, white) to 100 (variable, purple), shown in the same orientations as in (**a**). **c** Mapped footprints of antibodies AR4A (purple), HEPC74 (green), and collectively of antibodies that target the neutralizing face (gray), antigenic domain C (salmon), and antigenic domain A (pink), shown in same orientations as in (**a**) with N-linked glycans colored cyan. **d** Contribution of E2 N-linked glycan N695 (cyan) to the interface with antibody AR4A, with closeup view shown. **e** Contribution of E2 N-linked glycans N430 and N448 to the interface with HEPC74, with closeup view shown. HC heavy chain, LC light chain, AD antigenic domain. Source data are provided as a Source Data file.

the AR4A antibody targets an epitope on E2 with exceptionally low sequence variation across HCV genotypes. The high sequence conservation of the AR4A epitope likely reflects the structural importance of the bridging domain in E1E2 heterodimer integrity and possibly other functional roles that are yet to be defined. The mechanism of AR4A neutralization has been postulated to involve stabilization of the prefusion state of the glycoprotein or inhibition of conformational changes necessary for membrane fusion, although such a conclusion is difficult to confirm without knowledge of the full array of conformational states of E1E2, including those of putative fusogenic domains that remain disordered in existing structures[7].

**E1E2 glycosylation**

In the present structure of sE1E2.SZ, density corresponding to 9 predicted N-linked glycans within E2 was observed at positions N423,

N430, N448, N532, N540, N556, N576, N623, and N645, although to differing levels (Fig. 6). Density corresponding to a non-canonical N-linked glycan at E2 residue N695, which is conserved in >98% of reference genotype sequences, was also observed, confirming its occupancy in the genotype 1b context as well (Figs. 6 and 2b). Within E1, density corresponding to 4 out of 6 N-linked glycans was detected at positions N196, N209, N250, and N305, with N196 and N305 contributing to the E1E2 interface as noted above (Figs. 6, 3b, c, and 4d).

Examination of the observed N-linked glycans across the sE1E2.SZ structure revealed that they mainly clustered on one face of the heterodimer, spanning a semi-continuous surface from the E2 core to the E2 bridging domain and across to the E1 subunit (Fig. 6a–c)[6]. Strikingly, a 180° rotation of the structure revealed a face of the 1b09 E1E2 heterodimer that was mostly free of glycans, both within E1 and E2 (Fig. 6a–c). Unlike the majority of HCV genotypes, isolate 1b09 lacks an

N-linked glycan sequon at position 476 within the E2 VR2, which falls within the glycosylated face of the heterodimer (Fig. 6a, c). The glycan face of E1E2 in the context of other HCV genotypes is therefore likely even more densely glycosylated than what is observed here for 1b09 (Figs. 6a, c and 4e).

In view of the dichotomy in the density of glycans across these two faces of E1E2, we examined the degree to which each face was conserved in sequence across HCV genotypes. Mapping of Shannon sequence entropies of E1E2 surface residues revealed that the glycosylated face possesses a higher degree of variability than the non-glycosylated face (Fig. 6b). As such, a moderately conserved face of E1E2 is present across HCV genotypes but is not protected from immune recognition by a glycan shield. This face is also immunogenic but appears to be mainly targeted by non-neutralizing antibodies such as those that target antigenic domain A proximal to the bridging domain (Fig. 6c). We cannot exclude the possibility that portions of the non-glycosylated face of E1E2 may be partially protected from immune recognition by other means, such as at an interface within a putative higher-order oligomeric state or possibly through proximity to the viral membrane[8].

### E1E2 antigenic boundaries

Previous crystal structures of the E2 core defined a non-neutralizing face based on its lack of neutralizing antibody recognition, which in the present structure coincided with the non-glycosylated face[6]. To further assess how the observed N-linked glycosylation pattern on E1E2 correlated with known antigenic targets on E2, we mapped the footprints of two of the bound antibodies, AR4A and HEPC74, onto the E1E2 surface alongside previously defined antigenic targets on E2 (Fig. 6c). AR4A targets the E2 bridging domain and back layer while HEPC74 targets the E2 neutralizing face and prevents CD81 receptor recognition. The footprints of AR4A and HEPC74 were, in both cases, found in close proximity to E2 glycans and, in some cases, mediated direct interfaces with glycans (Fig. 6c–e). AR4A bound in proximity to N-linked glycan N695, while HEPC74 bound in close proximity to N-linked glycans at positions N430 and N448 (Fig. 6d, e). Roughly 10 and 23% of the paratope buried surface areas of the E2-AR4A and E2-HEPC74 interfaces involved glycans, respectively, although it is not clear whether glycan interactions influence antibody binding or are a secondary feature of antibody penetration of the glycan shield.

Collectively mapping the footprints of other antibodies that target the E2 neutralizing face revealed a high degree of overlap with the footprint of HEPC74, and thus proximity to glycans as well (Fig. 6c). The footprint of antibody HEPC46 that overlaps antigenic domain C also mapped in proximity to a glycan, in this case to N540 (Fig. 6c). Mapping the footprint of antibody 2A12 within antigenic domain A, placed it on the non-glycosylated face of E2 in proximity to the bridging domain. We note that 2A12 formed clashes with the bridging domain, likely a result of its induction by a truncated E2 antigen that lacked a bridging domain[5].

### Structural comparison with non-engineered full-length E1E2

A critical aspect of the present study was the use of a soluble secreted form of the E1E2 heterodimer that was engineered by replacing native transmembrane domains of the E1 and E2 subunits with a SYNZIP leucine zipper scaffold. While this design platform has been found to retain E1E2 antigenic integrity both in vitro and upon immunization and indeed has aided the production of soluble E1E2 for structural analysis, the question nonetheless existed as to whether the engineered aspects of sE1E2.SZ, in some way, led to structural divergence from non-engineered forms of the heterodimer[42,59]. While this manuscript was in preparation, a cryo-EM structure of a non-engineered membrane-extracted genotype 1a E1E2 heterodimer in complex with antibodies AR4A, AT1209, and IGH505 was reported (PDB 7T6X), which enabled comparison against the sE1E2.SZ structure[7]. Notably,

the membrane-extracted structure also utilized the AR4A antibody as a hook for purification of E1E2 and was complexed with an E2 neutralizing face-directed antibody, AT1209, and an antibody that targets the E1 C-terminal loop, IGH505. While the distinct genotypes used in each structure reflected amino acid sequence differences of 22.8% and 19.6% in their respective E1 and E2 ectodomains (Supplementary Fig. 8d), structural alignment of the two E1E2 structures revealed they were highly homologous, with a Cα r.m.s.d. of 1.34 Å (Supplementary Fig. 8a). Alignment of the individual E1 and E2 subunits from each structure, furthermore, yielded Cα r.m.s.d. values of 1.01 Å and 0.91 Å, respectively (Supplementary Fig. 8b,c). The structural divergence between the two structures was found between E2 residues 474–483 in VR2, which in the context of genotype 1a included an N-linked glycan sequon at position 476 that was absent in the genotype 1b sequence of sE1E2.SZ (Supplementary Fig. 8d). In addition, stem regions of E1 and E2, spanning residues 313–349 and 705–714, respectively, were poorly ordered in the sE1E2.SZ structure, but partially present in the membrane-extracted form. Overall, the comparison of the sE1E2.SZ structure against the non-engineered membrane-extracted E1E2 structure revealed remarkable homology, despite genotypic amino acid sequence divergence. Thus, the appended SYNZIP scaffold within sE1E2.SZ preserved the overall architecture of a native E1E2 ectodomain in the absence of membrane association, and sequence differences between genotypes 1a and 1b did not greatly impact the structure of E1E2.

## Discussion

The membrane-associated E1E2 surface glycoprotein of HCV is the main target of neutralizing antibodies during infection and is of central focus for vaccine development. While neutralizing antibodies have been identified that target individual E1 and E2 subunits, antibodies with extraordinary breadth of neutralization have been identified that depend on intact E1E2 heterodimers for recognition, including AR4A and AR5A, among others[18,22,60]. Vaccine antigens that retain native E1E2 antigenic features may thus hold quaternary epitopes that can facilitate induction of broadly neutralizing antibodies, although the design of such antigens has proven challenging due to the inherent flexibility of the glycoprotein and its membrane association[61–63]. Our team recently developed sE1E2 ectodomains that retained native antigenic features of the heterodimer by substitution of the E1 and E2 transmembrane domains with a human c-Fos/c-Jun leucine zipper scaffold[42]. We subsequently made a similar scaffolded E1E2 antigen, sE1E2.SZ, which replaced the human origin leucine zipper with a modified synthetic SYNZIP scaffold to minimize potential immunogenic complications related to scaffold-directed immune responses in humans, responses that can likely be further minimized by approaches that shield or occlude the scaffold.

In the present study, we utilized single particle cryo-EM to determine a 3.65 Å structure of the engineered sE1E2.SZ ectodomain of genotype 1b in complex with Fabs of neutralizing antibodies AR4A, HEPC74, and IGH520. The determined structure of sE1E2.SZ resolved the overall configuration of an E1E2 heterodimer liberated from its native transmembrane domains. A bridging domain at the C-terminus of E2 accounted for more than 60% of the interface with E1, while the N-terminus of E1 and its C-terminal loop mediated the majority of interactions with E2, including substantial contributions from two E1 N-linked glycans. Sequence analysis across a panel of reference HCV isolates revealed that residues of the E1E2 interface were highly conserved across HCV genotypes, as was the epitope of bound heterodimer-specific antibody AR4A that fell exclusively on E2. While HCV belongs to the *Flaviviridae* family of viruses, the E1E2 structure did not conform to structures of other flavivirus surface glycoproteins, such as Dengue or Zika, consistent with a lack of sequence homology[5,6]. We note that C-terminal stem regions of E1 and E2 and their associated SYNZIP scaffolds were not fully resolved in the

structure, nor was bound IGH520 Fab or regions of E1 previously implicated in membrane fusion and viral entry, making it difficult to ascertain the structural basis for these functions of the glycoprotein[13,64–66].

A noted feature of the 1b09 sE1E2.SZ structure was the clustering of N-linked glycans on one face of the heterodimer. While a glycan face was initially observed in the context of structures of the E2 core alone, the present structure, as well as that of full-length E1E2, revealed that the shield extends to the bridging domain of E2 as well as to the E1 subunit[6,7]. Our sequence analysis furthermore suggests the non-glycosylated face of the heterodimer is more conserved across HCV genotypes than the glycosylated face, possibly implicating its presence in an as yet undetermined interface. Mapping of the footprints of antibodies AR4A and HEPC74 (along with other neutralization face-targeting antibodies) revealed the antibodies bound in close proximity to glycans and, in some cases, directly interacted with them, although the relevance of glycan-bnAb interactions in the context of viral clearance and vaccine development remains an active area of investigation.

A cryo-EM structure of a membrane-extracted E1E2 heterodimer of genotype 1a was recently reported in complex with a set of antibodies similar to that used in the present study, including antibody AR4A, a neutralization face antibody AT1209, and an E1-specific antibody IGH505[7]. Since it was of a full-length form of E1E2, with intact transmembrane domains, the structure revealed the overall native assembly of a non-engineered E1E2 bound by antibodies. We therefore utilized the full-length E1E2 structure to assess whether the structure of engineered sE1E2.SZ retained native structural features of the glycoprotein. Comparison of the two structures revealed remarkable homology between their respective E1E2 ectodomains, despite diverging in amino acid sequence by ~20% due to their distinct genotypes. sE1E2.SZ thus retained native structural features of E1E2 and the SYNZIP scaffold did not impede, and likely promoted, the adoption of a native E1E2 structural configuration. It is possible that the observed structural homology between sE1E2.SZ and full-length E1E2 was due to a similar set of bound antibody ligands. However, the very recognition of sE1E2.SZ by such conformation-specific antibodies, ones induced in natural infection and that must recognize native E1E2 glycoproteins to mediate virus neutralization, provides further evidence that sE1E2.SZ preserves native features of the ectodomain. In addition, sE1E2.SZ is recognized by E1E2 heterodimer-specific antibodies that target antigenic regions that are distinct from AR4, namely AR5, further suggesting it possesses native features of E1E2 that are independent of recognition by any one particular antibody. We note as well that mutagenesis studies indicate a 1:1 correspondence between sE1E2.SZ and full-length membrane-extracted E1E2[42,59]. Taken together, our results show that sE1E2.SZ is a close structural mimic of the native form of the E1E2 heterodimer. The prospect of a soluble E1E2 ectodomain antigen such as sE1E2.SZ for vaccine development significantly simplifies the pipeline for antigen production, formulation, and quality control, as compared to membrane-extracted forms, including as a building block for higher order assemblies such as trimers of E1E2 heterodimers or nanoparticles.

## Methods

### sE1E2.SZ construct
The sE1E2.SZ construct was designed using the E1E2 sequence from the HCV genotype 1b isolate 1b09 54-v03. The transmembrane domains of E1 and E2 were substituted with modified versions of SYNZIP2 and SYNZIP1, respectively, with N-terminal fusion sites on SYNZIP1 and SYNZIP2 selected by comparison of the SYNZIP1/SYNZIP2 crystal structure (PDB 3HE5) with the Fos/Jun leucine zipper structure (PDB 1FOS), to maintain a helical register with the residues used in the sE1E2.LZ design[42]. This resulted in SYNZIP1 residue Leu 2 and SYNZIP2 residue Arg 2 (based on PDB 3HE5 residue numbering) being used as

N-terminal residues for E2 and E1 fusion, respectively. As with the sE1E2.LZ design, a three-residue spacer (sequence: PGG) was included between the E1 or E2 ectodomain C-terminus and the respective SYNZIP scaffold N-terminus. A 6X arginine spacer separated the E1-SYNZIP2 sequence from the E2-SYNZIP1 to allow efficient cleavage when co-transfected with a furin encoding plasmid. A C-terminal 6X histidine tag was added to the E2-SYNZIP1. Gene synthesis of this design was performed by Genscript, Piscataway, NJ.

### Antibody production
Expi293F cells or Expi293F GnTI- Cells (Thermo Fisher Scientific; catalog #s A14527 and A39240) grown in Expi293 expression medium were transiently transfected with plasmids encoding a 6x-His-tagged fragment of the antigen binding of the heavy chain and the light chain of either HEPC74 or IGH520. Cell culture supernatants were harvested 72 h post-transfection by centrifugation followed by filtration. Clarified supernatants were applied to cOmplete His-Tag Purification resin (Roche Diagnostics), and captured antibody Fabs were eluted according to the manufacturer's instructions. The eluted fractions were then subjected to size exclusion chromatography (SEC) using a Superdex 200 Increase 10/300 column (Cytiva Life Sciences). Eluted fractions corresponding to antibody Fabs were pooled and concentrated.

### sE1E2.SZ-AR4A ternary complex expression and production
Expi293F GnTI- HEK cells grown in Expi293 expression medium (Thermo Fisher Scientific) were transiently transfected with plasmids encoding 1b09 sE1E2.SZ, Strep-II-tagged AR4A Fab heavy chain, AR4A light chain, and furin using ExpiFectamine 293 transfection reagent (Thermo Fisher Scientific) according to manufacturer's instructions. Supernatants were harvested 72 h post-transfection and clarified using centrifugation. Clarified supernatants were filtered and applied to a Strep-Tactin XT 4Flow resin (IBA life sciences) column according to the manufacturer's instructions. Eluted fractions were then applied to a Superdex 200 Increase 10/300 size exclusion column and fractions corresponding to the sE1E2.SZ-AR4A ternary complex were pooled and concentrated. Mini-protean TGX stain-free gels (Bio-rad laboratories) were used for SDS-PAGE analyses to confirm the presence of complex components.

### Purification of quaternary and quinary complexes
Purified sE1E2.SZ-AR4A ternary complex was incubated with IGH520 Fab at threefold molar excess of Fab. The mixture was then subjected to SEC using a Superdex 200 Increase 10/300 column to separate the sE1E2.SZ-AR4A-IGH520 quaternary complex from unbound Fabs. The fractions corresponding to the quaternary complex were pooled, concentrated, and confirmed by SDS-PAGE analysis to contain quaternary complex components.

The purified quaternary complex was then incubated with HEPC74 Fab at a threefold molar excess of Fab, and the resulting quinary complex was then separated from unbound Fab by SEC, as described above. The purified sE1E2.SZ-AR4A-HEPC74-IGH520 quinary complex was confirmed by SDS-PAGE analysis to contain all complex components and was concentrated to ~1 mg/mL prior to cryo-EM grid preparation.

### Cryo-EM data collection, processing, and structure determination
The sE1E2.SZ-AR4A-HEPC74-IGH520 complex was imaged using an in-house Glacios 200 kV cryo-EM microscope (Thermo Fisher Scientific). Data were collected using SerialEM automated data collection software, and movies were recorded with a Gatan K3 camera[47]. Cryo-EM movies were patch motion corrected using cryoSPARC v2.15[48]. CTF parameters were estimated using the Patch CTF job within cryoSPARC. Blob picking was implemented on a small subset of the dataset, and the

resulting particles were used for 2D classification. Particles corresponding to 2D classes of the sE1E2.SZ-Fab complex were used for ab initio reconstruction, and the resulting volume was used to generate 2D templates for a template particle picking job applied to the full dataset, and 3,013,611 particles were selected. After selecting 2D classes consisting of sE1E2.SZ-Fab particles, 515,000 particles were subjected to multiple rounds of 3D classification in which classes containing the most density for E1 were selected. Then, 143,576 particles were then refined further using non-uniform refinement, which resulted in a primary cryo-EM map with a resolution of 3.65 Å as estimated by a gold-standard Fourier shell correlation curve using a threshold of 0.143.

In order to obtain improved IGH520 density, 2D and 3D classification was performed, favoring particles/volumes that contained clear density for the IGH520 Fab. The resulting particle stack consisting of 88,738 particles was used for local refinement targeting the heterodimer, HEPC74 and AR4A variable regions, and the IGH520 antibody. The primary map and the local refinement map were merged using Combine Focus Maps within the Phenix suite for model building and model refinement[67].

Iterative model building was performed using COOT with iterative rounds of real space refinement undertaken using the Phenix software suite[67,68]. Structural models were validated using Molprobity, and conformational validation of carbohydrate structures was undertaken using Privateer[69,70]. Molecular interfaces were assessed using PISA[71]. All figures depicting cryo-EM maps and models were generated using ChimeraX[72].

### Sequence analysis
Pre-aligned HCV reference genotype E1E2 sequences ($N = 187$) were downloaded from the Los Alamos National Laboratory HCV database[73], and the 1b09 E1E2 sequence was aligned to that set using the MAFFT multiple sequence alignment program[74]. Analysis of conservation and sequence entropy was performed using a custom Perl script. Raw entropies were calculated in units of nats. Scaled entropy values (0–100 scale) were linearly scaled based on a maximum Shannon entropy of 2.4 nats, which was approximately the maximum observed Shannon entropy of the E1E2 residues (versus the theoretical maximum for 20 amino acids of ~3.0 nats).

### AlphaFold modeling
Modeling of the 1b09 E1 and E2 ectodomains was performed using AlphaFold2[75] in the ColabFold Google Colab[49] in September 2022. AlphaFold2 uses deep learning to predict protein structures from sequences, achieving high predictive accuracy[75]. Default ColabFold options were used for the input processing and AlphaFold2 modeling, except structural refinement was selected ("use_amber" option) to correct minor geometric and structural errors in the output models, in accordance with the AlphaFold2 pipeline[75]. As specified by the default ColabFold parameters, no templates from previously determined structures were used in the AlphaFold2 modeling. The top-ranked ColabFold models for E1 and E2 were used for further analysis.

### Reporting summary
Further information on research design is available in the Nature Portfolio Reporting Summary linked to this article.

## Data availability
The cryo-EM data generated in this study have been deposited in the Electron Microscopy Data Bank under accession code EMD-29419. The atomic model generated in this study has been deposited in the Protein Data Bank under accession code 8FSJ. All other data supporting the findings of this study are available in the article and the Supplementary Information. Previously published structures used in the design of sE1E2.SZ are available under PDB accession codes 3HE5 and 1FOS. Previously published structures used for comparative analyses are available under PDB accession codes 7T6X, 4UOI, 6MEH, 6UYG, 6W03, 7MWX, 6MEJ, 6BKB, 6UYF, 6URH, 6UYM, 6BKD, 7JTG, 7JTF, 6WO4, and 7RFC. Source data are provided with this paper as a Source Data file.

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

## Acknowledgements

We thank Dr. S. Saif Hasan for advice related to cryo-EM data processing. This work was supported by NIH grants R01AI132212 to R.A.M., B.G.P., E.A.T., and T.R.F., R01AI168048 to R.A.M., B.G.P., E.A.T., T.R.F., and G.O., R21AI154100 to B.G.P., E.A.T., and T.R.F., AI158193 and AI168251 to M.L., and the University of Maryland Strategic Partnership: *MPowering the State* to T.R.F. and G.O.

## Author contributions

M.C.M., R.W., and J.D.G. expressed and purified proteins and performed biochemical assays. M.C.M., B.M.J., and E.P. collected and processed cryo-EM data. M.C.M., B.M.J., and G.O. built and refined the structural model. R.Y. and B.G.P. generated AI-predicted models and performed sequence analyses. M.C.M., B.M.J., E.A.T., B.G.P., R.A.M., T.R.F., and G.O. analyzed the data. M.L. provided unique reagents. B.G.P., T.R.F., and G.O. supervised research. M.C.M. and G.O. wrote the manuscript with contributions from other authors.

## Competing interests

The authors declare the following competing interests: B.G.P., T.R.F., E.A.T., and J.D.G. are co-inventors on a pending patent application (US application 18/252800) filed by the University of Maryland based on the secreted E1E2 design used in this study. T.R.F. is a co-founder and holds stock in NeuImmune, Inc., a company focused on glycoprotein-based vaccines and therapeutics. All the other authors declare no competing interests.
