## [Peer Review File · Nature Communications]

REVIEWER COMMENTS

Reviewer #1 (Remarks to the Author):

Summary:

Here the authors report the 4 angstrom resolution, cryo-EM reconstruction of a soluble HCV E1/E2 ectodomain heterodimer stabilized by a synthetic leucine-zipper in place of the transmembrane anchors of each. This structure is compared to previously reported structures of E1, E2, and the membrane associated E1/E2 heterodimer. Potential for vaccine development is invoked.

Major comments:

This manuscript confirms the previously published work by Torrents de la Pena et al. Science, however any new biology is unclear in this manuscript. Any unique features relative to the already published E1/E2 structure are not well presented here. The leucine zipper used to create the heterodimer is not ordered and so not included. There are no proposed or tested hypotheses based on the structure.

Please include the data not shown in the supporting materials (e.g. SDS-PAGE and SEC of the purified complex). Lastly, reference to supplementary methods is made but not included in submission. Perhaps I missed it.

E2 residues 644–700 have already been defined as the base domain, not the bridging domain, by Torrents de la Pena et al. Science.

Minor comments:

Line 27: single pass is hyphenated.

Ext data figures 4, 5, and 7 fail to identify models. This could be done as in Ext. data figure 6 and legend by the judicious addition of the word gray.

Reviewer #2 (Remarks to the Author):

In this study, Metcalf, et al. determined a cryo-EM structure of engineered soluble Hepatitis C virus glycoprotein E1E2 heterodimer in complex with Fab fragments from three neutralizing antibodies. Given the difficulties of expressing and purifying soluble E1E2 heterodimers, developing a robust system for the expression and purification of HCV E1E2 heterodimers is an unmet need in the HCV field. In their previous work, authors expressed and purified HCV E1E2 heterodimer using the Fos/Jun leucine zippers (LZ) that facilitated the heterodimerization of E1E2 glycoproteins (Guest, et al., 2021). In a current study, authors used synthetic analogs of Fos/Jun LZ to express, purify, and structurally characterize E1E2 in complex with Fabs. The authors performed a structural analysis of the E1-E2 and E1/E2-NAb interface. They also compared the E1E2_SZ structure with previously published structures of membrane-associated E1E2 in complexes with Fabs and previous structures of E2-Fab complexes.

The significance of this study appears rather limited due to the recently published structure of membrane-associated E1E2 in complex with three neutralizing Fabs (Torrents de la Pena et al, 2022). For E1E2 stabilization, both groups used Fabs that bind to the same antigenic regions/epitopes: (i) AR4A that binds to the E1E2 interface; (ii) HEPC74 (Metcalf, et al., current study) or AT1209 (Torrents de la Pena et al, 2022) that both bind to AR3 epitope in the front layer of E2; and (iii) IGH520 (Metcalf, et al, current study) or IGH505 (Torrents de la Pena et al, 2022) that both bind to E1 protein. It is unclear whether the high similarity of both structures is the result of Fab stabilization of flexible epitopes considering previous observations of intrinsic flexibility of E2 ectodomain. It would be more convincing if authors determined and characterized the structure of E1E2 in complex with antibodies that bind to different E1E2 epitopes.

Most of the manuscript focuses on characterizing E1E2_SZ heterodimer structure and Fab interactions with E1E2. However, since both engineered soluble E1E2 and previously published membrane-associated E1E2 structures are similar (Extended Data Figure 6), and both groups used Fabs that bind to the same epitopes, the structural information is limited to mainly the confirmation that soluble E1E2 heterodimer mimics membrane-associated E1E2, which is a significant achievement by itself.

Finally, the authors propose that the soluble E1E2 heterodimers characterized in this study might be used as vaccine candidates. However, there is no data in the manuscript that would indicate how homogeneous E1E2_SZ samples are before and after gel filtration chromatography. A previous study of E1E2_SZ precursor E1E2_LZ (Guest, et al., 2021) indicated the presence of high-order disulfide-linked aggregates after the affinity-column purification. It's unclear why the authors omitted gel filtration

traces and protein gel images before and after complexing E1E2_SZ with Fabs. Such information should be included to evaluate the usefulness of E1E2_SZ for future vaccine studies.

Minor comments

1. The supplementary methods section referenced several times was not available for review
2. Line 244: please check the numbering, L100d is likely to be L100e and F100e is likely to be F100d
3. Extended Data figure 1: Panels (e) and (f) need to be labeled in the figure
4. Extended Data figure 4: the color scheme needs to be described in the legend
5. Line 333: Antibody AT1209, used in for the crystallization of membrane-associated E1E2 (Torrents de la Pena et al., 2022), binds to the AR3 epitope, not the E1 protein. IGH505 antibody from that study binds to the E1 protein.

Reviewer #3 (Remarks to the Author):

In this work, Metcalf and colleagues present the structure of an engineered secreted ectodomain variant of the E1E2 surface glycoprotein from hepatitis C virus (HCV) in complex with three neutralizing antibodies, using cryo-electron microscopy (cryo-EM). In this E1E2 complex, the transmembrane helices of E1 and E2 are each replaced by a synthetic leucine zipper (SYNZIP) to stabilize a soluble E1E2 heterodimer. The cryo-EM structure of the soluble E1E2 variant shows a very high degree of similarity with the recently determined cryo-EM structure of membrane associated E1E2 (published during preparation of this manuscript, Torrents De La Peña et al., Science, 2022), thus preserving essential structural features of native E1E2, and can aid in the rational design of secreted E1E2 variants for vaccine development. Overall the manuscript presents a nice piece of structural work and will definitely be of interest to the field.

For this review I am mainly providing a technical assessment of the cryo-EM work, which is overall nicely performed.

I have following remarks and questions, which I believe should be addressed.

Main Remarks:

- At the start of data processing (pipelines described in Extended Data Figures 1 and 2) the choice is made to remove 2D classes corresponding to quaternary complexes (containing E1E2 + Fabs HEPC74 and AR4A) and only retain particles belonging to the 2D classes that seemingly correspond to a

quinary/pentameric complex additionally containing Fab IGH520. However, why was it not tried to also include the quaternary classes to perform ab-initio reconstruction and ensuing heterorefinement & Non-Uniform (NU) refinement in an attempt to increase the resolution for the quaternary E1E2:HEPC74:AR4A complex? The conformational heterogeneity could then be sorted out using 3D classification/heterorefinement rather than at the 2D classification stage. Particles corresponding to quaternary and quinary complexes could also be used together in a local refinement focusing on the quaternary complex part.

- It is stated in the pipeline described in Extended Data Figure 1 that after heterorefinement, 'the class with superior density (middle) was selected for further refinement'. It is not obvious to me, based on the figure, why the right class was deemed to be inferior. While refinement of the middle class leads to a final map with a nominal resolution just below 4 Å, it looks to me that the right class excluded for further refinement has more signal for the missing Fab (IGH520). While refinement of the right class (containing less particles) will likely result in a slightly lower resolution, maybe it would be possible to get more info on the location of Fab IGH520. Was refinement of this class tried at all?

- In the pipeline for Extended Data Figure 2, Local refinement was performed on a NU refinement based on a selected class (middle class after heterorefinement). However, since the goal of this local refinement is to obtain more signal for Fab IGH520, why were the particles corresponding to the right class after heterorefinement not included together with the particles for the middle class?

- In Extended Data Figure 3, a lower resolution map is shown that was used to build an extended model including additional signal for a part of Fab IGH520. Is this the merged map (combining NU refinement and local refinement) described in the methods section? If yes, this should be explicitly mentioned in the figure.

- Which maps were provided in the EMDB deposition (EMD-29419)? I believe that in addition to the main map (NU-refinement?) and two half maps, it is necessary to also deposit the Local Refinement map, and the merged (composite) map, together with the extended model described in Extended Data Figure 3.

- Map-model FSC curves are missing in the FSC plots and should be added. Furthermore, the model resolution at a map-model FSC threshold of 0.5 should be provided in Extended Data Table 1, rather than the currently provided map-model FSC threshold of 0.143 (Henderson & Rosenthal, Journal of Molecular Biology, 2003, Rosenthal & Rubinstein, Current Opinion in Structural Biology, 2015).

Minor remarks

Main text:

- All instances of 'electron density' should be replaced (p. 7, line 121 line 131, p. 8, line 140, 148, p. 9 line 179, p. 10 line 184, 193, p. 14 line 272, 274, 276, and maybe some I have missed), since a cryo-EM map is a Coulomb potential map and not an electron density map. 'Electron density map' can be replaced by cryo-EM map, Coulomb potential map, 3D reconstruction, ... 'Resolvable electron density enabled generation of a structural model corresponding to ...' should be replaced by something like 'The final map was of sufficient quality to generate a structural model for ...'.

- Line 58: 'All of these features have hindered the immunogen design process as well as structure determination of the E1E2 heterodimer'. This sentence gives the misleading impression that the structure of E1E2 is not yet determined, while the membrane bound E1E2 structure is already available.

- Lines 52 – 54: 'Structural characterization of antigenic targets on E2 has provided a great deal of information ...', No reference is provided for this sentence.

- Line 140: 'Low resolution ... was also detected', here the authors need to clarify which map was used here (i.e. the merged 'composite' map).

- Line 165 – 166: 'Commencing with a hinge like loop ..., it was mostly made up of ...'. This sentence needs to be revised. I would suggest: 'These include a hinge like loop ..., which is mostly made up of ...' or something similar.

- Line 250: 'structural basis for breath of AR4A recognition': maybe replace by 'structural basis of broad AR4A recognition'.

Discussion:

- The prospect of a soluble E1E2 ectodomain antigen such as sE1E2.SZ for vaccine development is mentioned in the discussion. The potential immunogenicity of the synthetic leucine zipper (SYNZIP) tag should be briefly discussed here.

Extended Data:

- Currently the resolution is listed as 3.9 Å in Extended Data Table 1, while the resolution is actually 3.99 (or 4.0 when rounded). Either 3.99 or 4.0 needs to be listed.

- Extended Data Figure 1: Which tool was used to calculate the local resolution estimation?

- Extended Data Figure 3: A map colored to local resolution is missing in the pipeline.
- Extended Data Figures 1 & 3: The gold-standard FSC cut-off (0.143) should be explicitly mentioned in the legend (or written above the blue line on the gold-standard FSC plots).
- All figures: it is not mentioned in the manuscript or figure legends which software was used for generation of structural renders (I assume PyMOL?) or display of maps (Chimera/ChimeraX?).

Point-by-Point Response to Reviewer Comments

We thank the reviewers for providing insightful feedback and comments on the manuscript, which we have incorporated into the revised version. We address the reviewer comments on a point-by-point basis below (in blue text).

REVIEWER COMMENTS

Reviewer #1 (Remarks to the Author):

Summary:

Here the authors report the 4 angstrom resolution, cryo-EM reconstruction of a soluble HCV E1/E2 ectodomain heterodimer stabilized by a synthetic leucine-zipper in place of the transmembrane anchors of each. This structure is compared to previously reported structures of E1, E2, and the membrane associated E1/E2 heterodimer. Potential for vaccine development is invoked.

Major comments:

This manuscript confirms the previously published work by Torrents de la Peña *et al.* *Science*, however any new biology is unclear in this manuscript. Any unique features relative to the already published E1/E2 structure are not well presented here. The leucine zipper used to create the heterodimer is not ordered and so not included. There are no proposed or tested hypotheses based on the structure.

We concur with the reviewer that the structure of sE1E2.SZ holds a high degree of structural homology to the structure of non-engineered full-length E1E2 reported by Torrents de la Peña *et al.*, *Science*, 2022. However, we view the observed similarities between the two structures as a relevant finding of the manuscript, one that reveals that the membrane-liberated sE1E2.SZ molecule retains a native E1E2 conformation that isn't biased by the leucine zipper scaffold. The structure validates sE1E2.SZ as a soluble native mimic of E1E2 and reinforces its strengths as a tractable candidate for rational vaccine development.

We also note that one difference between the E1E2 construct used in our structure and that used by Torrents de la Peña *et al.* was that ours was of genotype 1b sequence (1b09) while theirs was of genotype 1a sequence (AMS0232). These two genotypes differ in amino acid sequence by 22.8% and 19.6% within the E1 and E2 ectodomains, respectively. To clarify these sequence differences, we have added a sequence alignment of the two ectodomains to Supplementary Fig. 8 of the revised manuscript.

Please include the data not shown in the supporting materials (e.g. SDS-PAGE and SEC of the purified complex).

We thank the reviewer for alerting us to the absence of these data from the supporting materials. In the revised version of the manuscript we have added a new "Supplementary Fig. 1" that

includes a full description of the experimental pipeline used to generate the purified pentameric quinary complex. The new figure includes sequential SEC chromatograms and SDS-PAGE analyses that reflect each step of the pipeline and confirms the presence of all components in the complex.

Lastly, reference to supplementary methods is made but not included in submission. Perhaps I missed it.

We thank the reviewer for bringing this error to our attention. The sole methods section in the manuscript is the main text “Methods” section. Any references to “Supplementary Methods” have been corrected or removed.

E2 residues 644-700 have already been defined as the base domain, not the bridging domain, by Torrents de la Pena et al. Science.

Although we are aware of this difference in nomenclature, we have left the “bridging domain” nomenclature in place because we view it as an apt descriptor of this domain based on its placement directly in between E1 and E2, serving the function of bridging the two subunits. However, we also explicitly state the alternative nomenclature in the text (line 159) and Fig. 2 legend.

Minor comments:

Line 27: single pass is hyphenated.

The suggested correction has been made and now reads “single-pass”.

Ext data figures 4, 5, and 7 fail to identify models. This could be done as in Ext. data figure 6 and legend by the judicious addition of the word gray.

We thank the reviewer for alerting us to this omission. We have updated the legends of these figures to include identifiers for all shown models.

Reviewer #2 (Remarks to the Author):

In this study, Metcalf, et al. determined a cryo-EM structure of engineered soluble Hepatitis C virus glycoprotein E1E2 heterodimer in complex with Fab fragments from three neutralizing antibodies. Given the difficulties of expressing and purifying soluble E1E2 heterodimers, developing a robust system for the expression and purification of HCV E1E2 heterodimers is an unmet need in the HCV field. In their previous work, authors expressed and purified HCV E1E2 heterodimer using the Fos/Jun leucine zippers (LZ) that facilitated the heterodimerization of E1E2 glycoproteins (Guest, et al., 2021). In a current study, authors used synthetic analogs of Fos/Jun LZ to express, purify, and structurally characterize E1E2 in complex with Fabs. The authors performed a structural analysis of the E1-E2 and E1/E2-NAb interface. They also compared the E1E2_SZ structure with previously published structures of membrane-associated E1E2 in complexes with Fabs and previous structures of E2-Fab complexes.

The significance of this study appears rather limited due to the recently published structure of membrane-associated E1E2 in complex with three neutralizing Fabs (Torrents de la Pena et al, 2022). For E1E2 stabilization, both groups used Fabs that bind to the same antigenic regions/epitopes: (i) AR4A that binds to the E1E2 interface; (ii) HEPC74 (Metcalf, et al., current study) or AT1209 (Torrents de la Pena et al, 2022) that both bind to AR3 epitope in the front layer of E2; and (iii) IGH520 (Metcalf, et al, current study) or IGH505 (Torrents de la Pena et al, 2022) that both bind to E1 protein. It is unclear whether the high similarity of both structures is the result of Fab stabilization of flexible epitopes considering previous observations of intrinsic flexibility of E2 ectodomain. It would be more convincing if authors determined and characterized the structure of E1E2 in complex with antibodies that bind to different E1E2 epitopes.

We agree with the reviewer that structures of sE1E2.SZ in complex with antibodies that target other antigenic regions are desirable. While we cannot exclude the possibility that the structural homology observed between sE1E2.SZ structure and the Torrents de la Peña *et al.*, *Science*, 2022 structure was due to the similar set of bound antibodies, the final observed similarity between them continues to support the notion that the sE1E2.SZ design preserves native structural features of E1E2 that enable its recognition by such conformation-specific neutralizing antibodies to begin with. Until additional structures of E1E2 heterodimers in complex with other antibodies are obtained, this will remain an open question. We also note (see last paragraph of the Discussion) that the sE1E2 proteins are recognized by E1E2-specific antibodies that target antigenic regions on E1E2 that are distinct from AR4, namely AR5, suggesting they possess native antigenic features of E1E2 that are independent of AR4A recognition.

Most of the manuscript focuses on characterizing E1E2_SZ heterodimer structure and Fab interactions with E1E2. However, since both engineered soluble E1E2 and previously published membrane-associated E1E2 structures are similar (Extended Data Figure 6), and both groups used Fabs that bind to the same epitopes, the structural information is limited to mainly the confirmation that soluble E1E2 heterodimer mimics membrane-associated E1E2, which is a significant achievement by itself.

We thank the reviewer for noting that the observed structural homology between sE1E2.SZ and membrane-associated E1E2 represents a significant achievement by itself. As noted above, it suggests that sE1E2.SZ maintains E1E2 in a native-like conformation and further validates such antigens as viable candidates for vaccine development.

We note that one difference between our E1E2 construct and that used by Torrents de la Peña et al., is that ours was of the genotype 1b (1b09) while theirs was of genotype 1a (AMS0232). The two genotypes differ in amino acid sequence by 22.8% and 19.6% within the E1 and E2 ectodomains, respectively. To further clarify these sequence differences, we have added a sequence alignment of the ectodomains of two genotypes to Supplementary Fig. 8 in the revised manuscript.

Finally, the authors propose that the soluble E1E2 heterodimers characterized in this study might be used as vaccine candidates. However, there is no data in the manuscript that would indicate how homogeneous E1E2_SZ samples are before and after gel filtration chromatography. A previous study of E1E2_SZ precursor E1E2_LZ (Guest, et al., 2021) indicated the presence of

high-order disulfide-linked aggregates after the affinity-column purification. It's unclear why the authors omitted gel filtration traces and protein gel images before and after complexing E1E2_SZ with Fabs. Such information should be included to evaluate the usefulness of E1E2_SZ for future vaccine studies.

We thank the reviewer for alerting us to the absence of these data from the supporting materials. In the revised version of the manuscript we have added a new "Supplementary Fig. 1" that includes a full description of the experimental pipeline used to generate the pentameric quinary complex, including sequential SEC chromatograms and SDS-PAGE analyses reflecting each step of the process. We would draw the reviewer's attention to the SDS-PAGE analyses in this new figure that show similar (albeit not identical) migration of the E2 subunit under reducing and non-reducing conditions. These gels, along with the SEC chromatograms, suggest that a sizable portion of sE1E2.SZ protein that was co-expressed and co-purified with AR4A Fab was not forming higher-order disulfide-linked aggregates. A more comprehensive study of sE1E2.SZ as a vaccine antigen, including biochemical characterization of its unbound state relative to that of sE1E2.LZ is in preparation for a separate manuscript.

Minor comments

1. The supplementary methods section referenced several times was not available for review

We thank the reviewer for bringing this error to our attention. The sole methods section in the manuscript is the main text "Methods" section. Any references to "Supplementary Methods" have been corrected or removed.

2. Line 244: please check the numbering, L100d is likely to be L100e and F100e is likely to be F100d

The numbering of L100e and F100d were indeed labeled incorrectly in the text and have been corrected accordingly.

3. Extended Data figure 1: Panels (e) and (f) need to be labeled in the figure

We thank the reviewer for alerting us to this omission. Labels have been added to all panels in the figure, which corresponds to "Supplementary Fig. 2" in the revised version of the manuscript.

4. Extended Data figure 4: the color scheme needs to be described in the legend

We thank the reviewer for alerting us to this issue. We have updated the legend of this figure to include identifiers for all the shown models.

5. Line 333: Antibody AT1209, used in for the crystallization of membrane-associated E1E2 (Torrents de la Pena et al., 2022), binds to the AR3 epitope, not the E1 protein. IGH505 antibody from that study binds to the E1 protein.

We thank the reviewer for alerting us to this confusion. This sentence was revised to more clearly state the epitopes of IGH505 and AT1209 and now reads: “Notably, the membrane-extracted structure also utilized the AR4A antibody as a hook for purification of E1E2, and was complexed with an E2 neutralizing face-directed antibody, AT1209, and an antibody that targets the E1 C-terminal loop, IGH505.”

Reviewer #3 (Remarks to the Author):

In this work, Metcalf and colleagues present the structure of an engineered secreted ectodomain variant of the E1E2 surface glycoprotein from hepatitis C virus (HCV) in complex with three neutralizing antibodies, using cryo-electron microscopy (cryo-EM). In this E1E2 complex, the transmembrane helices of E1 and E2 are each replaced by a synthetic leucine zipper (SYNZIP) to stabilize a soluble E1E2 heterodimer. The cryo-EM structure of the soluble E1E2 variant shows a very high degree of similarity with the recently determined cryo-EM structure of membrane associated E1E2 (published during preparation of this manuscript, Torrents De La Peña et al., Science, 2022), thus preserving essential structural features of native E1E2, and can aid in the rational design of secreted E1E2 variants for vaccine development. Overall the manuscript presents a nice piece of structural work and will definitely be of interest to the field. For this review I am mainly providing a technical assessment of the cryo-EM work, which is overall nicely performed.

We thank the reviewer for the positive evaluation of the structural work and for providing an in-depth technical assessment and insightful suggestions on the cryo-EM analysis.

I have following remarks and questions, which I believe should be addressed.

Main Remarks:

- At the start of data processing (pipelines described in Extended Data Figures 1 and 2) the choice is made to remove 2D classes corresponding to quaternary complexes (containing E1E2 + Fabs HEPC74 and AR4A) and only retain particles belonging to the 2D classes that seemingly correspond to a quinary/pentameric complex additionally containing Fab IGH520. However, why was it not tried to also include the quaternary classes to perform *ab-initio* reconstruction and ensuing heterorefinement & Non-Uniform (NU) refinement in an attempt to increase the resolution for the quaternary E1E2:HEPC74:AR4A complex? The conformational heterogeneity could then be sorted out using 3D classification/heterorefinement rather than at the 2D classification stage. Particles corresponding to quaternary and quinary complexes could also be used together in a local refinement focusing on the quaternary complex part.

We thank the reviewer for this excellent suggestion. We pursued the suggested course of action and retained 2D classes corresponding to quaternary complexes in addition to presumed quinary classes. As now outlined in Supplementary Fig. 2, we performed multiple rounds of *ab initio* reconstruction (n=3) followed by heterorefinement. Two of the final classes were pooled and further refined using non-uniform refinement resulting in a new primary non-uniform map. This approach yielded improved cryo-EM map GSFSC resolution of 3.65 Å (down from 3.99 Å of our previous maps). The improved map yielded qualitative improvements in residue side-chains and augmented density for some N-linked glycans, including at position N423 of E2 that was not

previously observed. The improved map did not however yield substantial structural changes in the backbone trace of our primary model. Nonetheless, in view of this improved primary cryo-EM map we have updated cryo-EM map representations in Figs. 1, 2, and 3 of the manuscript to reflect this new map, and have updated the description of the cryo-EM pipeline in Supplementary Fig. 2. We also pursued a local refinement strategy as described below.

- It is stated in the pipeline described in Extended Data Figure 1 that after heterorefinement, ‘the class with superior density (middle) was selected for further refinement’. It is not obvious to me, based on the figure, why the right class was deemed to be inferior. While refinement of the middle class leads to a final map with a nominal resolution just below 4 Å, it looks to me that the right class excluded for further refinement has more signal for the missing Fab (IGH520). While refinement of the right class (containing less particles) will likely result in a slightly lower resolution, maybe it would be possible to get more info on the location of Fab IGH520. Was refinement of this class tried at all?

Refinement of the excluded (right) class was previously performed, and resulted in a lower GSFSC resolution of 4.6 Å. The resulting map did not unfortunately provide sufficient information to accurately model IGH520 or its E1 epitope.

- In the pipeline for Extended Data Figure 2, Local refinement was performed on a NU refinement based on a selected class (middle class after heterorefinement). However, since the goal of this local refinement is to obtain more signal for Fab IGH520, why were the particles corresponding to the right class after heterorefinement not included together with the particles for the middle class?

We concur with this point and as suggested by the reviewer this formed part of our updated strategy for obtaining extended density for E1 and IGH520 (see Supplementary Fig. 3). Specifically, after multiple rounds of heterorefinement, the two 3D classes referenced by the reviewer (middle and right) were used for non-uniform refinement, which resulted in a map with higher resolution (GSFSC = 3.85 Å). Local refinement of this map resulted in a considerable gain in IGH520 signal. A map generated by merging this local refinement map with the updated primary map from Supplementary Fig. 2 is now shown in Supplementary Fig. 4. Although the signal for IGH520 increased, we were still unable to accurately build the structural model of the antibody but did build in a putative IGH520 E1 helical epitope.

We note that local refinement of the new primary map shown in Supplementary Fig. 2 yielded a map that was qualitatively inferior to the map described in Supplementary Fig. 3, and was therefore not pursued further.

- In Extended Data Figure 3, a lower resolution map is shown that was used to build an extended model including additional signal for a part of Fab IGH520. Is this the merged map (combining NU refinement and local refinement) described in the methods section? If yes, this should be explicitly mentioned in the figure.

The map shown in the original Extended Data Figure 3 was indeed the merged map described in the methods section. The legend of the corresponding figure in the revised manuscript,

Supplementary Fig. 4, has been updated to state the nature of the new map.

- Which maps were provided in the EMDB deposition (EMD-29419)? I believe that in addition to the main map (NU-refinement?) and two half maps, it is necessary to also deposit the Local Refinement map, and the merged (composite) map, together with the extended model described in Extended Data Figure 3.

The main map, and two half maps were originally the only maps deposited in the EMDB. In view of the reviewer's suggestions we have now updated the deposited maps, which include the following: 1. Primary non-uniform map and its half maps, 2. Locally refined map and its half maps, 3. Composite merged map of the primary non-uniform map and the locally refined map.

- Map-model FSC curves are missing in the FSC plots and should be added. Furthermore, the model resolution at a map-model FSC threshold of 0.5 should be provided in Extended Data Table 1, rather than the currently provided map-model FSC threshold of 0.143 (Henderson & Rosenthal, Journal of Molecular Biology, 2003, Rosenthal & Rubinstein, Current Opinion in Structural Biology, 2015).

We have updated Supplementary Table 1 to include model resolution using a map-model FSC threshold of 0.5. Additionally, we have included map-model FSC curves for all of the deposited maps including the primary non-uniform map (Supplementary Figure 2) and the locally refined map (Supplementary Fig. 3).

Minor remarks

Main text:

- All instances of 'electron density' should be replaced (p. 7, line 121 line 131, p. 8, line 140, 148, p. 9 line 179, p. 10 line 184, 193, p. 14 line 272, 274, 276, and maybe some I have missed), since a cryo-EM map is a Coulomb potential map and not an electron density map. 'Electron density map' can be replaced by cryo-EM map, Coulomb potential map, 3D reconstruction, ... 'Resolvable electron density enabled generation of a structural model corresponding to ...' should be replaced by something like 'The final map was of sufficient quality to generate a structural model for ...'.

We thank the reviewer for alerting us to these issues in terminology, and the manuscript has been corrected accordingly.

- Line 58: 'All of these features have hindered the immunogen design process as well as structure determination of the E1E2 heterodimer'. This sentence gives the misleading impression that the structure of E1E2 is not yet determined, while the membrane bound E1E2 structure is already available.

We thank the reviewer for alerting us to the inadvertent impression this sentence may have conveyed. In the revised version of the manuscript we have removed this sentence. Additionally, we have added a more extensive paragraph to the Introduction that discusses the membrane-extracted E1E2 protein, including critical findings related to its recent cryo-EM structure and background on its use as an immunogen in a human vaccine trial.

- Lines 52 – 54: ‘Structural characterization of antigenic targets on E2 has provided a great deal of information ...’, No reference is provided for this sentence.

We thank the reviewer for alerting us to this unintentional omission. Multiple references describing antibody-bound E2 structures have been added to this sentence.

- Line 140: ‘Low resolution ... was also detected’, here the authors need to clarify which map was used here (i.e. the merged ‘composite’ map).

As suggested by the reviewer, we have clarified that the map being referenced is the merged composite map. The sentence now reads: “Low resolution density within the extended merged composite map corresponding to bound antibody IGH520 Fab was detected, but was not of sufficient resolution to permit unambiguous structure determination (Supplementary Fig. 4).”

- Line 165 – 166: ‘Commencing with a hinge like loop ..., it was mostly made up of ...’. This sentence needs to be revised. I would suggest: ‘These include a hinge like loop ..., which is mostly made up of ...’ or something similar.

This sentence was revised as per the reviewer’s suggestion and now reads: “The E2 bridging domain within sE1E2.SZ exhibited a number of notable features, including a hinge-like loop between residues 646-652, an intra-domain disulfide bond between residues C652 and C677, and a non-canonical glycan at position N695 (Figs. 2b-d)”

- Line 250: ‘structural basis for breadth of AR4A recognition’: maybe replace by ‘structural basis of broad AR4A recognition’.

This sentence was revised and now reads: “To assess the structural basis for broad AR4A recognition, we calculated the mean Shannon sequence entropy across residues of the of the AR4A epitope on E2, weighted by residue buried surface area, and compared it against similar values for the epitopes of 13 other HCV E2-directed antibodies, including HEPC74 (Fig. 5e,f).”

Discussion:

- The prospect of a soluble E1E2 ectodomain antigen such as sE1E2.SZ for vaccine development is mentioned in the discussion. The potential immunogenicity of the synthetic leucine zipper (SYNZIP) tag should be briefly discussed here.

We agree with the reviewer that the potential immunogenicity of the synthetic leucine zipper scaffold should be discussed. We have therefore added the following sentence to the first paragraph of the Discussion: “Given its placement within exposed C-terminal domains of E1 and E2, it is possible that immune responses against the scaffold itself might arise, although these can likely be minimized by approaches that shield or occlude it.”

Extended Data:

- Currently the resolution is listed as 3.9 Å in Extended Data Table 1, while the resolution is actually 3.99 (or 4.0 when rounded). Either 3.99 or 4.0 needs to be listed.

The updated resolution of the primary non-uniform map in Supplementary Table 1 is now listed as 3.65 Å.

- Extended Data Figure 1: Which tool was used to calculate the local resolution estimation?

CryoSPARC local resolution estimation software was used to calculate local resolution estimates, and we now note this in the legends of Supplementary Figs. 2 and 3.

- Extended Data Figure 3: A map colored to local resolution is missing in the pipeline.

As suggested by the reviewer, we have added maps colored by local resolution estimates to the corresponding pipeline that is now shown in Supplementary Fig. 3 of the revised manuscript.

- Extended Data Figures 1 & 3: The gold-standard FSC cut-off (0.143) should be explicitly mentioned in the legend (or written above the blue line on the gold-standard FSC plots).

The gold-standard FSC cut-off (0.143) are now explicitly listed in the respective figures, and also mentioned in the corresponding legends.

- All figures: it is not mentioned in the manuscript or figure legends which software was used for generation of structural renders (I assume PyMOL?) or display of maps (Chimera/ChimeraX?).

Chimera X was used to render all figures, and we now note this in the methods section of the revised manuscript.

REVIEWERS' COMMENTS

Reviewer #3 (Remarks to the Author):

The Authors have adequately addressed all of my comments and suggestions. It is nice to see that the resolution and overall quality of the main cryo-EM map could be significantly improved in the revised manuscript. I have no further comments and recommend publication of the revised manuscript.

Best regards,

Jan Felix

Response to Reviewer Comments

We thank the reviewers for providing insightful comments and suggestions on the original version of the manuscript, which were incorporated into the revised version.

REVIEWERS' COMMENTS

Reviewer #3 (Remarks to the Author):

The Authors have adequately addressed all of my comments and suggestions. It is nice to see that the resolution and overall quality of the main cryo-EM map could be significantly improved in the revised manuscript. I have no further comments and recommend publication of the revised manuscript.

Best regards,

Jan Felix

We thank Dr. Jan Felix for providing an in-depth evaluation of the cryo-EM analysis and of the manuscript as a whole. In particular, the provided suggestions led to significant improvements in the resolution of the cryo-EM maps.